# Changes in BVOC emissions in response to the El Niño-Southern Oscillation

Ryan Vella[1,2], Andrea Pozzer[1,4], Matthew Forrest[3], Jos Lelieveld[1,4], Thomas Hickler[3,5], and Holger Tost[2]

[1]Atmospheric Chemistry Department, Max Planck Institute for Chemistry, Mainz, Germany
[2]Institute for Atmospheric Physics, Johannes Gutenberg University Mainz, Mainz, Germany
[3]Senckenberg Biodiversity and Climate Research Centre (SBiK-F), Frankfurt am Main, Germany
[4]Climate and Atmosphere Research Center, The Cyprus Institute, Nicosia, Cyprus
[5]Department of Physical Geography, Goethe University, Frankfurt am Main, Germany

**Correspondence:** Ryan Vella (ryan.vella@mpic.de)

**Abstract.**

Emissions of Biogenic volatile organic compounds (BVOCs) from the terrestrial biosphere play a significant role in major atmospheric processes. BVOCs are highly reactive compounds that influence the atmosphere's oxidation capacity and also serve as precursors for the formation of aerosols that influence global radiation budgets. Emissions depend on the response of vegetation to atmospheric conditions (primarily temperature and light), as well as other stresses, e.g., from droughts and herbivory. The El Niño–Southern Oscillation (ENSO) is a naturally occurring cycle arising from anomalies in the sea surface temperature (SST) in the tropical Pacific. ENSO perturbs the natural seasonality of weather systems on both global and regional scales and is considered the most significant driver of climate variability. Several studies have evaluated the sensitivity of BVOC fluxes during ENSO events using historical transient simulations. While this approach employs realistic scenarios, it is difficult to assess the impact of ENSO alone given the multiple types of climate forcing, e.g., from anthropogenic emissions of $CO_2$ and aerosol. In this study, a global atmospheric chemistry–climate model with enabled interactive vegetation was used to conduct two sets of simulations: 1) isolated ENSO event simulations, in which a single ENSO event is used to perturb otherwise baseline conditions, and 2) sustained ENSO simulations, in which the same ENSO conditions are reproduced for an extended period of time. From the isolated ENSO events, we present global and regional BVOC emission changes resulting from the immediate response of vegetation to atmospheric states. More focus is given to the sustained ENSO simulations, which have the benefit of reducing the internal variability for more robust statistics when linking atmospheric and vegetation variables with BVOC flux anomalies. Additionally, these simulations explore long-term changes in the biosphere with potential shifts in vegetation in this possible climate mode, accounting for the prospect of increased intensity and frequency of ENSO with climate change. Our results show that strong El Niño events increase global isoprene emission fluxes by 2.9% and that one single ENSO event perturbs the Earth system so markedly that BVOC emission fluxes have not returned to baseline emissions within several years after the event. We show that persistent ENSO conditions shift the vegetation to a new quasi-equilibrium state, leading to an amplification of BVOC emission changes with up to 19% increase in isoprene fluxes over the Amazon. We provide evidence that BVOC-induced changes in plant phenology, such as the leaf area index (LAI), have a significant influence on BVOC emissions in the sustained ENSO climate mode.

# 1 Introduction

The terrestrial biosphere is a major source of natural volatile organic compounds (VOCs), such as isoprene and monoterpenes, which account for approximately 90% of all VOC emissions to the atmosphere (Guenther et al., 1995). Isoprene and monoterpenes are thought to work against stress-induced reactive oxygen species or help plants in coping with abiotic stress by changing membrane properties (Sharkey and Loreto, 1993; Vickers et al., 2009; Karl et al., 2010; Sharkey and Monson, 2017), while they can also be induced by other chemical, physical, or biological processes, such as herbivory (Laothawornkitkul et al., 2008) and signaling between organisms (Zuo et al., 2019). Biogenic volatile organic compounds (BVOCs) are highly reactive and short-lived (minutes to hours) as they quickly interact with tropospheric oxidant gases upon emission, exerting a significant influence on the atmosphere's oxidation capacity (Atkinson, 2000; Atkinson and Arey, 2003). The dominant reaction mechanisms of BVOCs are OH and $O_3$ oxidation, which have significant implications for secondary organic aerosol (SOA) formation, and in turn, for cloud formation and climate (Pöschl et al., 2010; Ehn et al., 2014; Palm et al., 2018). OH oxidation of BVOCs also has implications for greenhouse gas and pollutant concentrations (e.g., methane and CO; Arneth et al., 2010; Peñuelas and Staudt, 2010). BVOC emissions exhibit year-to-year variations attributed to the vegetation's sensitivity to climatic conditions. Our prior investigations have revealed that isoprene fluxes, spanning a decade, display a standard deviation of 8 Tg yr$^{-1}$ (Vella et al., 2023).

The El Niño–Southern Oscillation (ENSO) is a periodic oscillation (occurring every 2 to 7 years) between anomalously warm (El Niño) and cold (La Niña) sea surface temperatures (SSTs) in the tropical Pacific (McPhaden et al., 2006). Because of the strong interactions between atmospheric and oceanic circulations, such anomalies in tropical Pacific SST have a significant impact on atmospheric processes. ENSO therefore exerts a marked influence on weather systems and global climate patterns (McPhaden et al., 2006). During ENSO events, the Walker circulation convective centers re-arrange, inducing precipitation anomalies in the tropics as well as influencing monsoon systems via the Hadley circulation over the Pacific, Indian, and Atlantic Oceans. Teleconnections with midlatitude westerlies can also result in consistent anomaly patterns in the extratropics (Dai and Wigley, 2000). To this end, tropical regions are often much warmer and drier than average during El Niño eposides (Gong and Wang, 1999; Dai and Wigley, 2000), but some regions tend to be cooler and wetter, e.g., western North America (Ropelewski and Halpert, 1986) and East Asia (Wu et al., 2003).

Several studies have explored the sensitivity of the terrestrial biosphere to different ENSO phases (e.g., Ahlström et al., 2015; Chang et al., 2017; Bastos et al., 2018; Wang et al., 2018; Teckentrup et al., 2021). The primary factors driving changes in vegetation are closely linked to the dominant meteorological drivers of net primary productivity (NPP) in different regions. Specifically, the primary drivers in the wet tropics and dry tropics and temperate regions are, respectively, radiation and moisture, while temperature is the primary driver in the western temperate and boreal regions (Nemani et al., 2003). The terrestrial biosphere often acts as a carbon source during El Niño events, while carbon uptake increases during La Niña events, particularly in semi-arid ecosystems (Ahlström et al., 2015). Nevertheless, the complex relationships between ENSO-induced climatic variability and terrestrial ecosystem productivity, particularly the extent, amplitude, and underlying processes, remain poorly understood (Gonsamo et al., 2016; Wang et al., 2018; Zhu et al., 2017). Zhang et al. (2019) linked ENSO seasonality with

global gross primary production (GPP) and found peak correlations occurring between global yearly GPP and ENSO conditions in August and October of the preceding year. Drought and warming in the Amazon basin during the 2015/16 El Niño event led to higher stress-related BVOC emissions, which, when combined with greater turbulent transport above the canopy, resulted in higher OH reactivity compared to non-El Niño years (Pfannerstill et al., 2018). Isoprene emissions based on satellite formaldehyde (HCHO) measurements also show interannual differences tied to temperature shifts and climate features such as El Niño (Wells et al., 2020). Modelling studies have suggested that BVOC emissions are generally higher during El Niño years and lower during La Niña years, with ENSO having a significant impact in both the tropics and the higher latitudes (Naik et al., 2004; Lathiere et al., 2006; Müller et al., 2008). However, uncertainties persist due to the influence of climate change on ENSO variability.

Despite the numerous studies focusing on the effects of ENSO perturbation on the biosphere, only a limited number of studies have specifically investigated the sensitivity of BVOC emissions. Given that *in-situ* observations of BVOC emission fluxes are scarce, many studies have relied on satellite-based remote sensing or modelling approaches. The modelling studies, in particular, are limited in number and only link BVOC emission trends during ENSO events without exploring in detail the underlying mechanisms driving such changes or separating the effects of ENSO from the transient emission changes in greenhouse gases and shorter-lived pollutants and their precursors. BVOC emissions are influenced by the overall response of the biosphere to atmospheric conditions; however, they are also strongly influenced by temperature and surface radiation, making them likely to be affected by the temperature and cloudiness variations associated with ENSO.

In this study, sea surface temperatures (SSTs) and sea ice cover (SICs) associated with ENSO years were used to construct different ENSO scenarios: very strong and moderate El Niño / La Niña scenarios as well as a baseline scenario. An Earth System Model (ESM) with interactive vegetation representations was used to investigate ENSO-induced changes in meteorology, vegetation, and BVOC emissions on global and regional scales. We present two sets of simulations: (1) isolated ENSO event simulations, which employ a single ENSO event in otherwise baseline conditions, and (2) sustained ENSO simulations, which use the same ENSO conditions over a 30-year period. Both sets of simulations are used to evaluate the various effects of ENSO. The isolated ENSO event simulations are used to study the temporal evolution of BVOC anomalies following a single ENSO event and to estimate global and regional changes resulting from the immediate response of the biosphere to atmospheric states anomalies. The most recent IPCC assessment based on CMIP6 simulations states that ENSO will continue to be the primary mode of interannual variability in a warmer climate, with a high degree of certainty (Lee et al., 2021). However, some studies have suggested the possibility of increased frequency of extreme ENSO events under greenhouse warming (e.g., Cai et al., 2015, 2021). Therefore, we use the sustained ENSO simulations to explore the upper range of impacts of ENSO, as well as long-term changes in the biosphere and the resulting BVOC emission fluxes in such a climate mode. We have examined various parameters, including surface temperature, surface radiation, aridity, net primary productivity (NPP), and leaf area index (LAI), to investigate the changes brought by ENSO. While changes in these parameters on a global scale can indicate overarching trends, anomalies connected to the ENSO are often identified at a regional level. Given that the majority of BVOC emissions take place in tropical regions, our analysis focuses on seven specific regions within or in close proximity to the tropics: South West USA (SWUSA), Amazon Basin (Amazon), Central West Africa (CEAfr), South East Africa (SEAfr), India, South East

Asia (SEAsia), and North Australia (NAus), and these regions are commonly thought to be hotspots for ENSO-associated
climate anomalies. To our knowledge, no study has comprehensively investigated global and regional BVOC emission changes
in relation to vegetation and atmospheric state anomalies elicited by ENSO. Therefore, this research aims to shed light on the
intricate interactions that drive shifts in BVOC emissions associated with the El Niño-Southern Oscillation.

## 2 Methods

### 2.1 Models

**The EMAC modelling system**

The EMAC (ECHAM/MESSy Atmospheric Chemistry) model is a numerical chemistry and climate modelling system that
contains submodels that represent tropospheric and middle atmospheric processes, as well as their interactions with oceans,
land, and anthropogenic activities. It originally combined the ECHAM atmospheric GCM (Roeckner et al., 2006) with the
Modular Earth Submodel System (MESSy) (Jöckel et al., 2005) framework and philosophy, modularizing physical processes as
well as most of the infrastructure into submodels that can be further developed to improve existing process representations; new
submodels can also be added to represent new or alternative process representations. In recent years, EMAC has been further
developed to include a broader representation of atmospheric chemistry by coupling different processes such as representations
for aerosols, aerosol–radiation and aerosol–cloud interactions (Tost, 2017). In this study, version 2.55 has been used, which is
based on the well documented version used in comprehensive model intercomparison studies (Jöckel et al., 2016).

**LPJ-GUESS**[1]

The Lund-Potsdam-Jena General Ecosystem Simulator (LPJ-GUESS) (Smith et al., 2001, 2014) is a dynamic global vegetation
model (DGVM) featuring an individual-based model of vegetation dynamics. These dynamics are simulated as the emergent
outcome of growth and competition for light, space, and soil resources among woody plant individuals and a herbaceous
understorey in each of a number (50 in this study) of replicate patches representing random samples of each simulated locality
or grid cell. The simulated plants are classified into 12 plant functional types (PFTs) discriminated by growth form, phenology,
photosynthetic pathway (C3 or C4), bioclimatic limits for establishment and survival and, for woody PFTs, allometry and life
history strategy. LPJ-GUESS has previously been implemented in global ESMs (e.g., Weiss et al., 2014; Alessandri et al.,
2017), and, more recently, coupled with EMAC (Forrest et al., 2020; Vella et al., 2023). The LPJ-GUESS version used in this
study currently provides information on potential natural vegetation, and it does not incorporate changes in land use. This is a
limitation within our current model configuration, as the implemented version of LPJ-GUESS lacks the capability to account
for land use changes. This functionality will be included in the next version of LPJ-GUESS.

---

[1]The following section is based on the template for standard copyright-free LPJ-GUESS model description (https://web.nateko.lu.se/lpj-guess/resources.
html,last access: 03 July 2023).

## 2.2 EMAC-LPJ-GUESS configuration

Once fully coupled, the EMAC-LPJ-GUESS configuration will be a sophisticated Earth system model (ESM) capable of studying interactions between the land and atmosphere. This includes examining the methane cycle and its lifespan, the atmospheric chemistry of various carbon compounds, the impact of fires and associated feedbacks, future nitrogen deposition rates and scenarios for fertilization, the effects of ozone on plants, and the role of biogenic volatile organic compounds on aerosol production and their contribution to cloud formation and precipitation patterns. While efforts towards a fully coupled configuration are ongoing, in this work, we use the standard EMAC-LPJ-GUESS coupled configuration, where the vegetation in LPJ-GUESS is entirely determined by the EMAC atmospheric state, soil, N deposition, and fluxes (Forrest et al., 2020), but there is no feedback from the vegetation to climate variables (e.g., changes in albedo and roughness length). After each simulation day EMAC computes the average daily values of 2-meter temperature, net downwards shortwave radiation, and total precipitation and passes these state variables to LPJ-GUESS. Vegetation information (LAI, foliar density, leaf area density distribution, and PFT fractional coverage) from LPJ-GUESS is then fed back to EMAC for the calculation of BVOC emission fluxes using EMAC's BVOC submodules (Vella et al., 2023), namely ONEMIS (Kerkweg et al., 2006) and MEGAN (Guenther et al., 2006). Both ONEMIS and MEGAN are based on the Guenther algorithms (Guenther et al., 1993, 1995), where the BVOC emission flux ($F$) is calculated as a function of the foliar density and its vertical distribution ($D$ [kg dry matter m$^{-2}$]), ecosystem-specific emission factors ($\epsilon$), and a non-dimensional activity factor ($\gamma$) that accounts for the photosynthetically active radiation (PAR) and temperature:

$$F = [D] \, [\epsilon] \, [\gamma] \tag{1}$$

In this work, we evaluate fluxes from ONEMIS, which is the standard and most established emission module in EMAC. Emissions are calculated at four distinct canopy layers, which are defined by the leaf area density (LAD) and the leaf area index (LAI). The attenuation of the PAR is determined for each level by considering the direct visible radiation and the zenith angle. Using the proportions of sunlit leaves and the overall biomass, emissions from both sunlit and shaded leaves within the canopy are estimated. Further technical details for canopy processes employed in ONEMIS can be found in Ganzeveld et al. (2002). While validating pure BVOC fluxes from models using observations remains challenging, this setup was evaluated and demonstrated to be adept in capturing global BVOC estimates and responses when compared to other modelling studies (Vella et al., 2023). As described in Eq. 1, BVOC emission calculations in this setup are governed by vegetation states ($D$) from LPJ-GUESS that are largely based on temperature, radiation, and soil moisture. Furthermore, the instantaneous surface radiation and temperature levels ($\gamma$) have a large impact on the emission rates. On the the basis of such model parameterisations, we explore the impact on BVOC emission anomalies by evaluating changes in the surface temperature and radiation, the aridity index (AI), the NPP, and the LAI.

## 2.3 Experimental design

A thirty-year (1980–2009) SST & SIC dataset from the Atmospheric Model Intercomparison Project Phase II (AMIP II) was used to evaluate the Oceanic Niño Index (ONI) and classify the strength of different ENSO events (https://pcmdi.llnl.gov/
mips/amip/amip2/, last access: 03 July 2023). Running 3-month mean SST anomalies in the Niño 3.4 region (5°N-5°S, 120°-170°W), based on this 30-year base period, are shown in the top panel of Fig. 1. Events are defined when the anomaly is greater or equal to $0.5°C$ for five consecutive overlapping 3-month periods ($+0.5°$ for El Niño events and $-0.5°$ for La Niña events). Furthermore, events are categorised into Weak (0.5 to 0.9 SST anomaly), Moderate (1.0 to 1.4), Strong (1.5 to 1.9), or Very Strong ($\geq 2.0$) when the threshold is reached or exceeded for at least three consecutive overlapping 3-month periods.
Even though not officially published, this ONI threshold classification has been used by the National Oceanic and Atmospheric Administration (NOAA)[2] and also in several research articles (e.g., Jimenez et al., 2021; Abish and Mohanakumar, 2013).

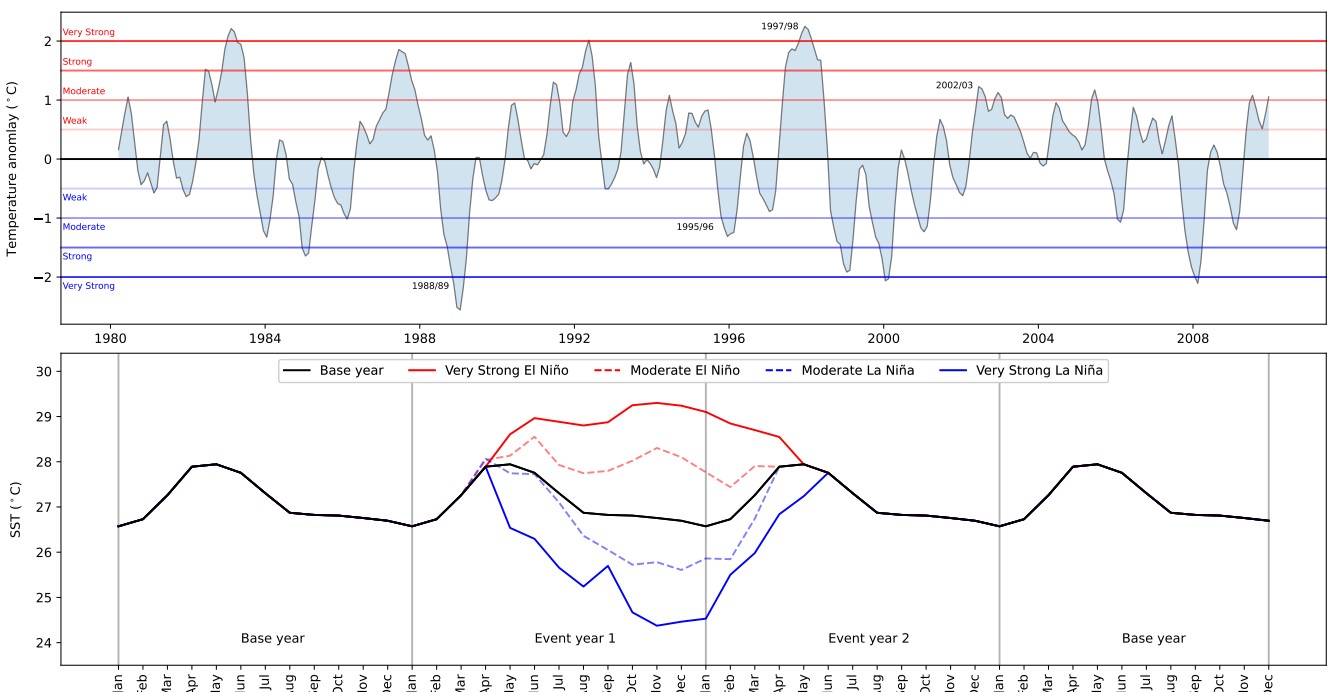

**Figure 1.** Top panel: SST anomalies in Niño 3.4 region from 1980 to 2009. Horizontal lines indicate the anomaly strength (El Niño in red & La Niña in blue) with incremental thresholds every $0.5°$. The dates of significant ENSO episodes considered in this study are highlighted. Bottom panel: Four consecutive years of SST in Niño 3.4 region. The two years in the middle indicate four ENSO scenarios as well as the base scenario. The base year (i.e., the 30-year average SST over El Niño 3.4) is shown in black, while the ENSO events are shown in red (El Niño) and blue (La Niña).

---

[2]www.climate.gov/news-features/blogs/enso/united-states-el-niÃśo-impacts-0, last access: 03 July 2023

Global SST and SIC are used as forcing data to construct five scenarios: (1) Base conditions (based on the 1980–2009 average); (2) Very Strong El Niño (based on 1997/98); (3) Moderate El Niño (based on 2002/03); (4) Very Strong La Niña (based on 1988/89); and (5) Moderate La Niña (based on 1995/96). The event typically spans from March to June of the following year. The lower panel of Fig. 1 shows SST values in the El Niño 3.4 region. During ENSO events, SST starts to deviate from baseline conditions in March/April of the first event year and continues until April/May/June of the following year (event year 2). For all simulations, the $CO_2$ concentration was kept fixed at 348 ppmv, representing the year 2000. This study looks at atmospheric, vegetational, and BVOC emission changes in sustained ENSO scenarios with continuous ENSO forcing, as well as the temporal evolution of BVOC emissions following an isolated event under otherwise baseline conditions. To do so, two sets of global simulations, hereinafter referred to as *sustained* and *isolated* simulations, are performed. Both sets of simulations run at a horizontal resolution of T63 (approximately $1.9° \times 1.9°$) and have a 500-year offline spin-up phase. In this work, we focus especially on seven regions, defined in the supplementary material and also presented graphically in Fig. 4, 5, and 7. The regions considered are hotspots for ENSO (apart from NAus) and places with generally high BVOC emissions in the tropics (except from SWUSA) (Bastos et al., 2013; Vella et al., 2023; Sindelarova et al., 2014). Additionally, we used the BVOC anomaly distribution maps (Fig. 7) to establish the exact dimensions of the bounding box for regions with relatively consistent BVOC anomalies. Throughout our analysis, we applied an ocean mask to focus solely on anomalies occurring over land.

**Isolated ENSO event simulations**

This simulation setup is designed to trigger a single ENSO event (spread over two years) in otherwise baseline conditions. The evolution of the SST temporal simulations is depicted in the lower panel of Fig. 1. The simulations used base conditions for the whole simulation time except for the two event years, where a perturbation in the SST and SIC is introduced according to the specific ENSO event considered. The ENSO anomaly perturbation is employed in the 31[st] and 32[nd] years of the 50-year simulation, which means that the simulation runs with base SST/SIC from the 1[st] to the 30[th] year, and then from the 33[rd] to the 49[th] year.

The events considered with this setup are Very Strong La Niña event (based on SST and SIC from May to December 1988 and January to April 1989), and Very Strong El Niño event (based on SST and SIC from May to December 1997 and January to March 1998). With this setup, temporal variations in BVOC emissions during Very Strong ENSO events and in subsequent years can be assessed. We also investigate correlations between the BVOC flux anomalies and the temperature, radiation, aridity index (AI), net primary production (NPP), and leaf area index (LAI) anomalies during the event years and the subsequent two years. Here, the AI is defined as the total precipitation divided by the potential evaporation/sublimation (including evapotranspiration (PET)). The standardized anomaly is calculated by dividing the anomalies by the standard deviation of the base scenario. The simulations conducted in this study differ from previous studies, where BVOC emission changes due to ENSO are typically evaluated using satellite data (e.g., Zhang et al., 2019; Wells et al., 2020) or with transient (i.e., historical) simulations (e.g., Naik et al., 2004; Bastos et al., 2018; Teckentrup et al., 2021). The advantage of conducting isolated ENSO simulations with constant climate boundary conditions is that it allows for the study of the specific impacts of ENSO on the

system being simulated. By isolating the ENSO signal and keeping other climatic forcing factors, such as trends from $CO_2$, aerosol loading, etc., constant, we can attribute any observed changes in BVOC fluxes solely to the ENSO phenomenon.

### Sustained ENSO simulations

Even though the isolated ENSO simulations give insights into changes in BVOC fluxes with respect to the magnitude and evolution over time, it is hard to constrain ENSO-induced changes statistically from a single simulation run given the high internal variability within the system. We therefore further analysed model results from sustained ENSO simulations. These simulations describe ENSO conditions continuously over many years by using the same yearly cycle of SST and SIC data over 30 simulated years. The five scenarios employ the following global SST and SIC data: (1) Base simulation - monthly average SST and SIC from 1980 to 2009 (30 years); (2) Moderate El Niño - April to December 2002 and January to March 2003; (3) Very Strong El Niño - May to December 1997 and January to March 1998; (4) Moderate La Niña scenario - April to December 1995 and January to March 1996; and (5) Very Strong La Niña scenario - May to December 1988 and January to April 1989. The corresponding SSTs used in these simulations can be seen in the lower panel of Fig. 1. The baseline simulation employs SST/SIC spanning from January to December, whereas the ENSO simulations adopt distinct 12-month sequences spanning the event years 1 and 2 (see Fig. 1). In these subsequent simulations, there could be a disruption in the sequential order of months given that the ENSO event occurs over two years and could start in March, April, or May of the first event year. Nevertheless, the annual ENSO cycle remains consistent given that these 12 monthly SST/SIC data are used perpetually.

Compared to the isolated simulations, the sustained simulations better constrain the correlations between BVOC flux emissions, meteorology, and vegetation changes, and they provide statistical confidence that the characterised perturbations are caused by ENSO rather than other variability attributed to the climate system. However, we emphasise that these simulations express ENSO scenarios where the vegetation comes into quasi-equilibrium with the new climate system. The reported changes in BVOC emissions are therefore exaggerated as they include drifts in the vegetation states resulting from years to decades of plant establishment and mortality. The focus here is to link ENSO with the driving variables for BVOC emissions; however, given the possible increased frequency of more intense ENSO with climate change (Cai et al., 2021, 2015) these simulations provide insights on possible vegetation changes and the subsequent effects into BVOC emission fluxes in these scenarios.

| Simulation Name | Simulation Type | Details |
|---|---|---|
| Base<br>Very Strong El Niño<br>Very Strong La Niña | Isolated ENSO simulations | T63 horizontal resolution.<br>500-year offline spin-up phase.<br>40 years long, using the last 11 years for analysis.<br>Base global SST & SIC conditions except for the "event" years<br>on the 31$^{st}$ / 32$^{nd}$ year of the simulation. |
| Base<br>Very Strong El Niño<br>Moderate El Niño<br>Very Strong La Niña<br>Moderate La Niña | Sustained ENSO simulations | T63 horizontal resolution.<br>500-year offline spin-up phase.<br>50 years long, using the last 30 years for analysis.<br>Same conditions applied in every year of the simulation. |

**Table 1.** Simulation details for ENSO experiments in this study.

## 2.4 Principle Component Analysis

Principal Component Analysis (PCA) was used on climate variables to assess their correlation with isoprene emissions during El Niño events (Section 3.2.3). PCA is a statistical method used to simplify and understand complex datasets by transforming the original variables into a new set of orthogonal (uncorrelated) variables called principal components. These components are linear combinations of the original variables that capture the maximum variance in the data. For each pixel of the model output, we perform PCA to extract the first principal component for each driving variable (temperature, radiation, aridity index, NPP, and LAI). The correlation between the first principal component of each driving variable and isoprene/monoterpene emission fluxes is computed for each pixel. These correlation values are then used to rank the variables' importance in driving isoprene/monoterpene emissions during El Niño events for each pixel.

## 3 Results

### 3.1 Isolated ENSO events simulations

This section presents results from the isolated simulations described in Section 2.3. Fig. 2 shows global 12-month moving averages of monthly isoprene emissions globally and over seven regions (see Table 1 in supplementary material). The event, spanning over two years, is marked in green, while the following two years are marked in yellow. Isoprene fluxes from El Niño, La Niña, and base conditions are shown in red, blue, and black, respectively. Notice that prior to the event, the isoprene fluxes in all cases are identical; however, following the ENSO perturbation, fluxes diverge. The anomalies in BVOC emission fluxes result from the immediate response of vegetation due to changes in meteorological states elicited by ENSO. Given that the simulations are identical, the changes noted here can be solely attributed to the corresponding SST/SIC anomalies, and the influence of other forcing factors within the Earth system can be excluded.

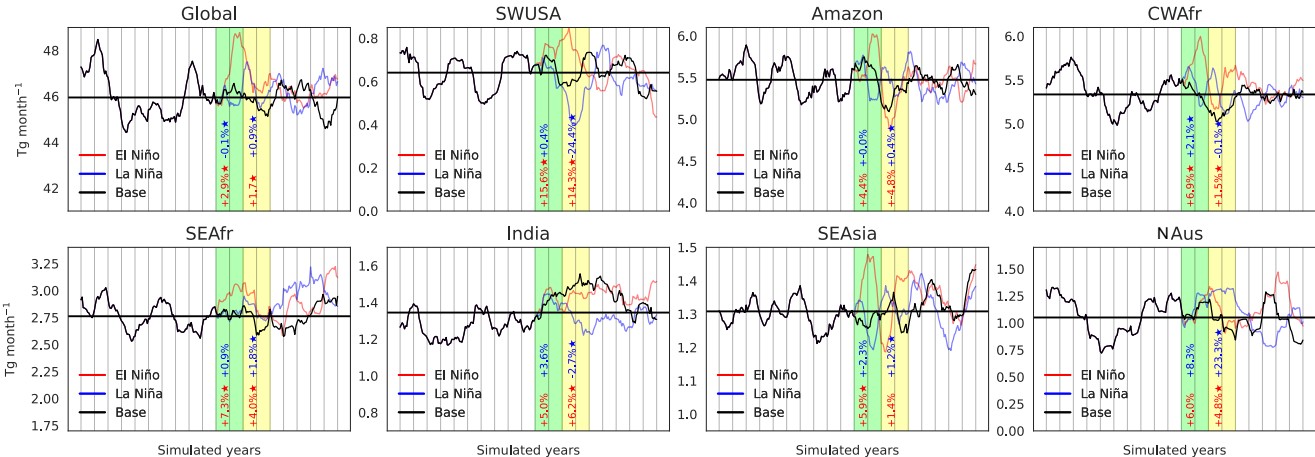

**Figure 2.** Evolution over time of global and regional isoprene fluxes in response to El Niño (red) and La Niña (blue) events over two years (green columns). The black line shows fluxes without the ENSO perturbation, while the horizontal black line illustrates the mean over base conditions throughout all simulations. Percentage changes of El Niño (in red) and La Niña (in blue) fluxes compared to the base simulation during the event (in green) and the subsequent two years (in yellow) are included with statistically significant changes (*p*<0.01) marked with a star. Vertical grey lines correspond to simulated years. Only 10 of the 30 simulated years before the ENSO event are shown.

Fig. 2 shows percentage changes of El Niño (in red) and La Niña (in blue) fluxes compared to the base simulation during the
event (in green) and the subsequent two years (in yellow). The percentage changes include a star symbol when the difference from the base emissions is statistically significant (99% confidence with a two-tailed Student's t-test, i.e., $p < 0.01$), over that time-frame. Global isoprene emissions increase by 2.9% during an El Niño and remain elevated by 1.7% in the two years following the event. During El Niño and the subsequent two years, SWUSA experiences a rise of 15.6% and 14.3%, respectively, while a decline of 24.4% is found in SWUSA during the two years following La Niña. Other notable emission
changes were observed in CWAfr, SEAfr, and SEAsia for the El Niño event. A significant increase of 23.3% in the two years following La Niña was also observed in NAus. Our results suggest that changes in isoprene fluxes due to an ENSO event are highly regional and, on a global scale, higher emissions during and following a strong El Niño event are present. These simulations also show that the perturbation introduced by the ENSO event stretches for a long time following the event, even though the SST and SIC are restored to Base conditions. Over the course of seven years following the event, global isoprene
emissions deviated from baseline levels by 0.74 Tg month$^{-1}$ during El Niño and 0.37 Tg month$^{-1}$ during La Niña on average. However, our findings indicate that the maximum monthly deviation reached 1.01 Tg during El Niño and 1.49 Tg during La Niña within this seven-year period. Monoterpene emissions follow similar tendencies as isoprene fluxes (see Fig. S3 in the supplement). El Niño results in a global increase of 3.2% in monoterpene fluxes, while regionally, emissions increase by 18.4% over SWUSA, 6.9% over the Amazon, 6% over CWAfr, 5.1% over SEAfr, and 2.6% over SEAsia ($p < 0.01$). La Niña events
do not result in statistically significant changes in monoterpene emission changes globally, but notable changes are seen in the two years following the event over SWUSA ($-24.2\%$), and NAus (+ 24.0%).

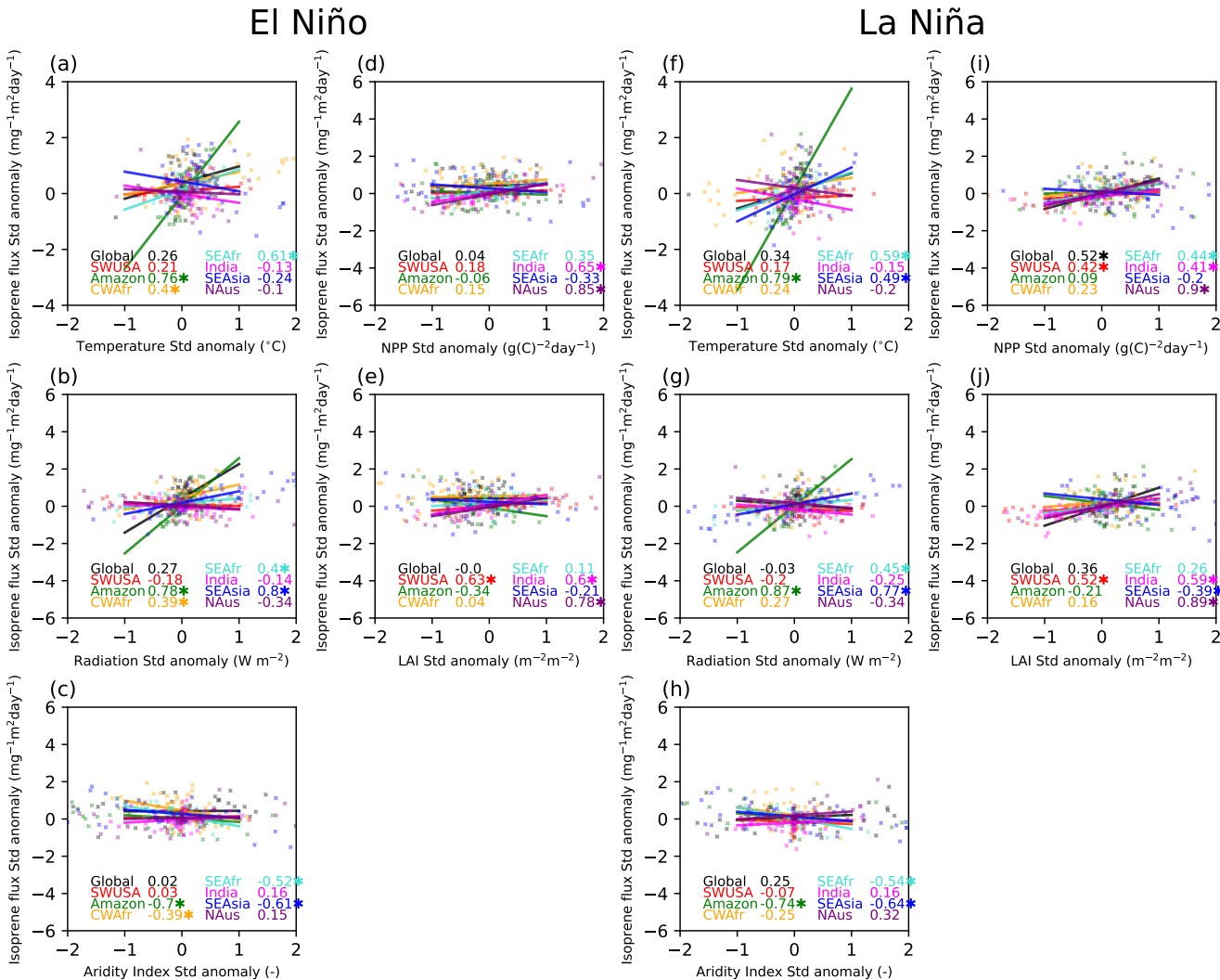

**Figure 3.** Scatter plots with a linear regression fit including the Pearson correlations (*r*) between the standardized isoprene flux anomalies and standardized temperature, radiation, aridity index (AI), net primary production (NPP), and leaf area index (LAI) anomalies in different regions for the event years and the following two years (four years in total). Very Strong El Niño events are shown in the panels on the left hand side while Very Strong La Niña events are shown in the panels on the right hand side. Correlations with $p < 0.01$ are marked with a star sign.

Fig. 3 shows correlations between the isoprene standardized flux anomalies, and the standardized temperature, radiation, aridity index (AI), net primary production (NPP), and leaf area index (LAI) anomalies during the event years and the following two years (total of four years) for the regions considered, during El Niño and La Niña. Table 2 lists the Pearson correlation

coefficients (*r*) from Fig. 3 but also includes *r* values for the 2-year event period, and the following two years separately. In our assessment, we classify the Pearson correlation coefficients as follows: 0.00-0.29 as *negligible*, 0.30–0.49 as *weak*, 0.50–0.69

as *moderate*, 0.70–0.89 as *strong*, and $\geq 0.90$ as *very strong* for positive correlations and similarly for negative correlation between 0 and $-1$. The investigated correlation coefficients between modelled isoprene emission fluxes and various driving variables provide insights into the relationship between these factors in different regions during El Niño and La Niña events.

| El Niño | Temperature | | | Radiation | | | AI | | | NPP | | | LAI | | |
|---|---|---|---|---|---|---|---|---|---|---|---|---|---|---|---|
| | Both | Event | +2 years | Both | Event | +2 years | Both | Event | +2 years | Both | Event | +2 years | Both | Event | +2 years |
| Global | 0.26 | 0.22 | 0.25 | 0.27 | 0.27 | 0.25 | 0.02 | 0.12 | −0.03 | 0.04 | −0.39 | **0.57** | **0** | −0.17 | 0.18 |
| SWUSA | 0.21 | 0.20 | 0.35 | −0.18 | −0.17 | −0.34 | 0.03 | 0.19 | 0.04 | 0.18 | 0.25 | −0.07 | **0.63** | 0.5 | **0.68** |
| Amazon | **0.76** | **0.89** | **0.53** | **0.78** | 0.73 | **0.85** | **−0.70** | **−0.77** | **−0.56** | −0.06 | −0.23 | 0.27 | −0.34 | **−0.68** | 0.28 |
| CWAfr | **0.40** | **0.51** | −0.07 | **0.39** | 0.58 | 0.1 | **−0.39** | −0.46 | −0.32 | 0.15 | 0 | 0.43 | 0.04 | −0.02 | 0.11 |
| SEAfr | **0.61** | **0.61** | **0.61** | **0.4** | 0.3 | **0.53** | **−0.52** | −0.45 | **−0.56** | 0.35 | 0.28 | **0.54** | 0.11 | 0.23 | 0.04 |
| India | −0.13 | −0.26 | −0.04 | −0.14 | −0.4 | 0.14 | 0.16 | 0.35 | 0.02 | **0.65** | **0.72** | 0.2 | **0.60** | **0.59** | 0.19 |
| SEAsia | −0.24 | −0.2 | **0.72** | **0.80** | **0.87** | **0.81** | **−0.61** | **−0.76** | −0.38 | −0.33 | **−0.62** | −0.01 | −0.21 | **−0.59** | −0.15 |
| NAus | −0.10 | 0.18 | −0.23 | −0.34 | −0.28 | −0.36 | 0.15 | 0.32 | 0.09 | **0.85** | **0.85** | **0.87** | **0.78** | **0.79** | **0.81** |

| La Niña | Both | Event | +2 years | Both | Event | +2 years | Both | Event | +2 years | Both | Event | +2 years | Both | Event | +2 years |
|---|---|---|---|---|---|---|---|---|---|---|---|---|---|---|---|
| Global | 0.34 | **0.44** | 0.27 | −0.03 | −0.25 | 0.13 | 0.25 | 0.27 | 0.22 | **0.52** | 0.42 | **0.59** | 0.36 | 0.08 | **0.61** |
| SWUSA | 0.17 | 0.24 | 0.01 | −0.20 | −0.09 | −0.47 | −0.07 | −0.09 | 0.17 | **0.42** | 0.26 | 0.51 | **0.52** | 0.45 | 0.48 |
| Amazon | **0.79** | **0.85** | **0.72** | **0.87** | **0.87** | **0.87** | **−0.74** | **−0.75** | **−0.72** | 0.09 | −0.15 | 0.33 | −0.21 | −0.22 | −0.17 |
| CWAfr | 0.24 | 0.45 | 0.14 | 0.27 | 0.3 | 0.35 | **−0.25** | −0.32 | −0.19 | 0.23 | 0.15 | 0.38 | 0.16 | 0.2 | −0.04 |
| SEAfr | **0.59** | **0.63** | **0.61** | **0.45** | 0.15 | **0.71** | **−0.54** | −0.40 | **−0.68** | **0.44** | 0.41 | 0.46 | 0.26 | 0.42 | 0.11 |
| India | −0.15 | −0.07 | −0.08 | −0.25 | −0.29 | 0.01 | 0.16 | 0.35 | 0.02 | **0.41** | **0.72** | 0.2 | **0.59** | **0.59** | 0.19 |
| SEAsia | **0.49** | **0.71** | **0.62** | **0.77** | **0.84** | **0.72** | **−0.64** | **−0.62** | **−0.67** | −0.20 | **−0.49** | −0.01 | **−0.39** | **−0.59** | −0.15 |
| NAus | −0.20 | −0.20 | −0.1 | −0.34 | −0.44 | −0.25 | 0.32 | **0.58** | 0.11 | **0.90** | **0.93** | **0.88** | **0.89** | **0.90** | **0.87** |

**Table 2.** The Pearson correlations ($r$) between the standardized isoprene flux anomalies and standardized temperature, radiation, aridity index (AI), net primary production (NPP), and leaf area index (LAI) anomalies at different regions for both events as well as the two subsequent years (as depicted in Fig. 3), for the ENSO event only (two years) and for the two years following the event. Correlations with $p < 0.01$ are in bold.

We note that the dependencies in isoprene emissions are very region-specific, so much so that correlations on a global scale are generally poor for both El Niño and La Niña events. In the Amazon region, during both El Niño and La Niña events, there are moderate-to-strong positive correlations observed between isoprene flux and temperature (El Niño = 0.89, La Niña = 0.85), as well as surface radiation (El Niño = 0.73, La Niña = 0.87). Additionally, there are strong negative correlations between isoprene emissions and the AI (El Niño = −0.77, La Niña = −0.75). These findings suggest that higher temperatures and surface

radiation tend to increase isoprene emissions in this region, while increased drought conditions limit the emissions to some degree. In CWAfr we see a moderate-to-weak correlations between the isoprene flux and temperature (0.51), radiation (0.58), and AI (-0.46) during an El Niño event and only a weak correlation with temperature (0.45) during a La Niña event. In SEAfr we also observe a moderate correlation with temperature (El Niño = 0.61, La Niña = 0.63), and a weak negative correlation with AI (El Niño = −0.45, La Niña = −0.40). In SEAsia, we observe a strong positive correlation (0.71) between temperature and isoprene flux during La Niña events. Additionally, there are strong correlations with radiation during both El Niño (0.87)

and La Niña (0.84) events. In this region, we also find weak-to-moderate negative correlations between isoprene emissions and the NPP and LAI for both El Niño and La Niña events. In India, the correlation between isoprene flux and temperature is negligible-to-weak across all time frames, suggesting that temperature alone may not be a strong driver of isoprene emissions in India; however, moderate-to-strong correlations are seen with AI, NPP, and LAI, during La Niña. In SWUSA there are

moderately positive correlations between isoprene flux and NPP/LAI during La Niña. Overall, the correlations in the NAus region suggest that surface the NPP, and LAI play significant roles in driving isoprene emission fluxes.

## 3.2 Sustained ENSO simulations

The isolated ENSO simulations discussed so far give insights into the temporal evolution of changes in BVOC emission changes from single ENSO events. Here we present results from sustained ENSO conditions based on the last 30 ensemble
285 years from 50-year simulations at a horizontal resolution of T63 with a 500-year offline spin-up phase. These simulations have the advantage of minimising the influence of internal variability and thus enhancing potential links between ENSO, meteorology, vegetation, and BVOC emission fluxes.

While the current climatic conditions do not align with this scenario, it has been suggested that under the influence of climate change, both El Niño and La Niña could become more intense and prolonged (Cai et al., 2015, 2021). The main objective of the
290 presented simulations is to statistically analyse the response of the driving variables to ENSO, enabling us to examine potential lasting impacts on the biosphere and BVOC emissions.

### 3.2.1 Changes in atmospheric drivers during ENSO

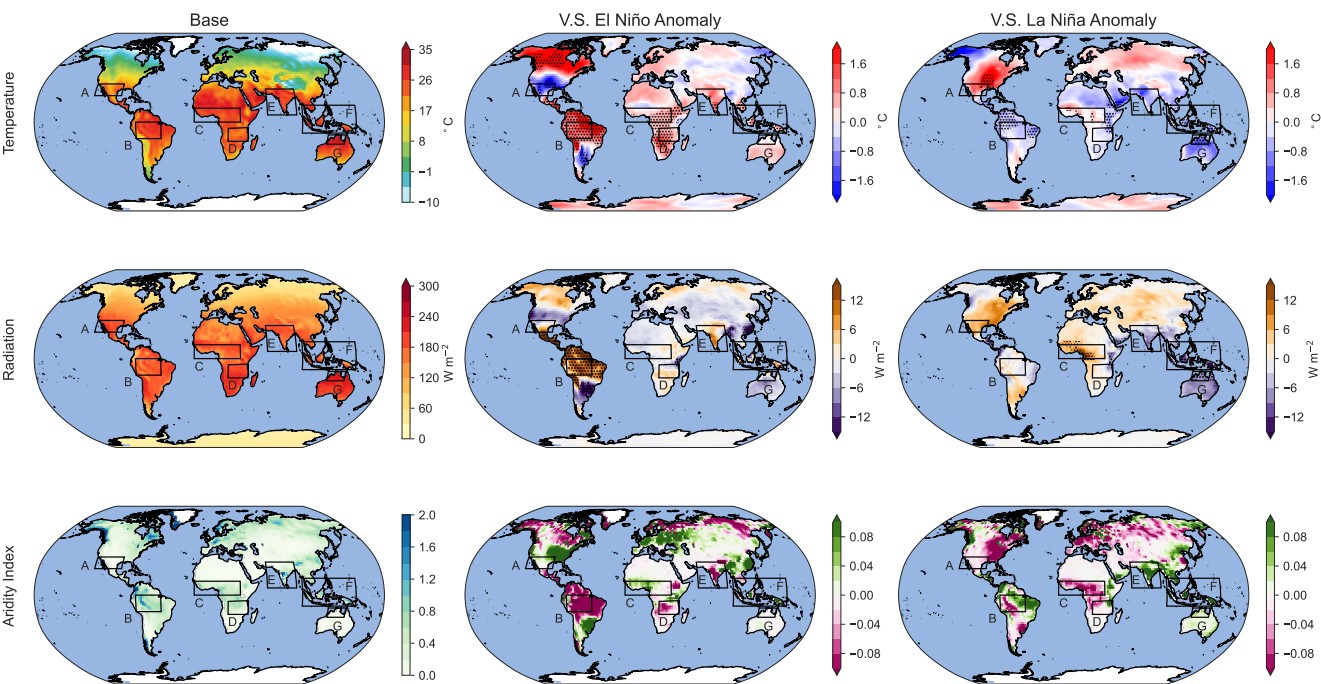

**Figure 4.** Global distribution of surface temperature, solar radiation, and AI during the base scenario (left panels) averaged over the last 30 years of 50-year simulations. The middle panels show El Niño anomalies and the right panels show LA Niña anomalies. Areas with statistically significant anomalies ($0 < .01$) are hatched.

Fig. 4 shows global distributions of surface temperature, net solar radiative flux at the surface, and the AI averaged over 30 years for the base scenario as well as anomalies from Very Strong El Niño (Very Strong El Niño − Base) and La Niña (Very

Strong La Niña − Base) scenarios. Table 3 provides an overview of the absolute and percentage changes in surface temperature, radiation, and AI over land during the ENSO conditions compared to the base scenario, globally, and the indicated areas of interest with statistically significant anomalies ($p < 0.01$) in bold.

| Area | El Niño Anomaly | | | La Niña Anomaly | | |
|---|---|---|---|---|---|---|
| | Temp [% (°C)] | Rad [% (W m$^{-2}$)] | AI [% (-)] | Temp [% (°C)] | Rad [% (W m$^{-2}$)] | AI [% (-)] |
| Global | 2.11 (0.25 °) | 0.49 (0.72) | −0.34 (−0.0008) | −0.26 (−0.03°) | 0.04 (0.06) | −0.79 (−0.0019) |
| SWUSA | −3.93 (−0.81°) | 0.02 (0.05) | 57.57 (0.027 ) | 3.74 (0.77 °) | 3.05 (5.96) | −47.23 (−0.0222) |
| Amazon | **5.84(1.51°)** | **6.02**(10.61) | −22.59 (−0.123) | **−1.49 (−0.39°)** | −0.12 (−0.22) | 5.45 (0.0297) |
| CWAfr | **2.35**(0.64°) | 3.44 (6.85) | −13.64 (−0.0442) | **−0.33 (−0.40°)** | **−2.71(2.56)** | 10.80 (0.035) |
| SEAfr | **2.80**(0.65°) | 0.03 (0.07) | −4.54 (−0.0126) | −1.21 (−0.28 °) | −0.82 (−1.60) | 2.47 (0.0069) |
| India | 2.44 ( 0.62 °) | 1.45 (2.72) | −11.67 (−0.0187) | −1.75 (−0.45 °) | −2.89 (−5.42) | 17.22 (0.0276) |
| SEAsia | **0.62(0.17°)** | 6.29 (12.91) | −20.04 (−0.1043) | **−0.42(−0.12°)** | **−3.36 (−6.89)** | 29.35 (0.1527) |
| NAus | 0.36 (0.11 °) | −0.53 (−1.26) | 2.55 (0.0007) | **−2.55(−0.76°)** | **−3.51(−8.28)** | 82.74 (0.0236) |

**Table 3.** Absolute and percentage changes in temperature, radiation, and the aridity index (AI) for El Niño and La Niña scenarios corresponding to Fig. 4. Statistically significant anomalies ($p < 0.01$) are in bold.

On a global scale, there is a notable increase in surface temperature over land (2.11%) during El Niño and a decrease (−0.26%) during La Niña. Global surface radiation anomalies and AI anomalies are below 1%. The SWUSA experiences a decrease in temperature (−3.93%) during El Niño, coupled with an increase in AI (57.57%). Conversely during La Niña, SWUSA encounters a temperature increase (3.74%) along with an increase in radiation (3.05%), but a substantial decrease in AI (−47.23%). The Amazon region exhibits a statistically significant increase in temperature (5.84%) and radiation (6.02%) during El Niño, while the aridity index undergoes a considerable decrease (−22.59%). For La Niña, the Amazon region experiences a statistically significant decrease in temperature (−1.49%) and an increase of 5.45% AI. In CWAfr, El Niño brings about an increase in temperature (2.35%), while AI decreases (−13.64%). During La Niña, CWAfr sees a decrease in temperature (−0.33%), a decrease in radiation (−2.71%) and an increase in AI (10.80%). SEAfr experiences a statistically significant increase in temperature (2.80%) during El Niño, while La Niña brings a decrease (−1.21%). Changes in the surface radiation and are relatively small for SEAfr, while the AI decreases (−4.54%) and increases (2.47%) during El Niño and La Niña, respectively. India experiences a temperature increase (2.44%) during El Niño and a decrease during La Niña (−1.75%). The AI shows contrasting changes, decreasing during El Niño (−11.67%) and increasing during La Niña (17.22%). SEAsia exhibits a slight temperature increase (0.62%) and a significant increase in radiation (6.29%), and a decrease in AI (−20.04%) during El Niño. Conversely, La Niña brings a slight temperature decrease (−0.42%), a decrease in radiation (−3.36%), and a increase in AI (29.35%). NAus experiences slight increase (0.36%) in temperature during El Niño, while temperature (−2.55%) and radiation (−3.51%) decrease during La Niña. The AI increases by 82.74% during La Niña. Overall, both El Niño and La Niña substantially modulate key meteorological variables at a regional level without disturbing the global energy and water budgets.

### 3.2.2 Vegetation response

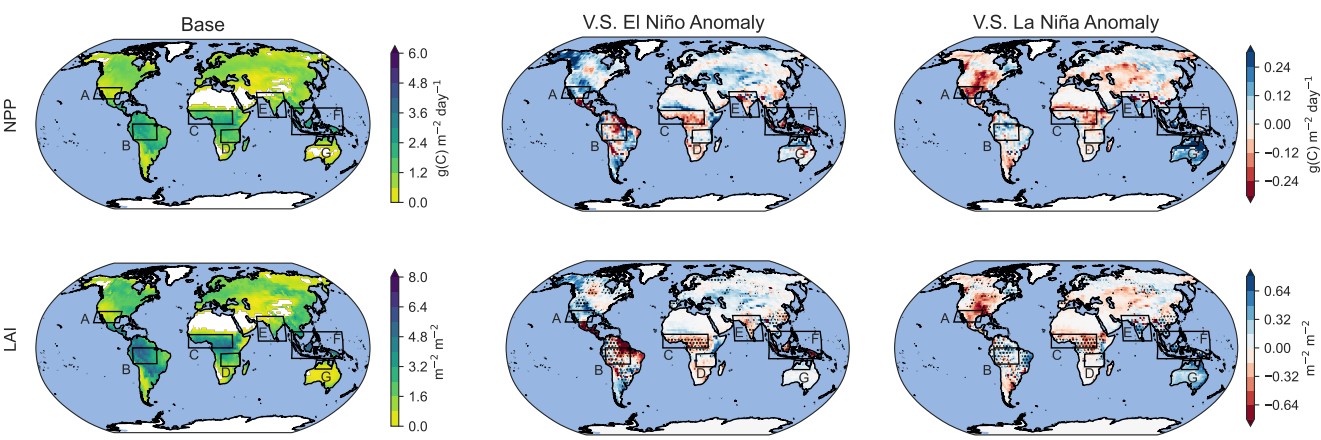

**Figure 5.** Distribution of NPP and LAI for base conditions (left panels) and anomalies in Very Strong El Niño (middle panels) and Very Strong La Niña (right panels) averaged over the last 30 years of 50-year simulations.

|  | El Niño Anomaly | | La Niña Anomaly | |
|---|---|---|---|---|
| Area | NPP [% (g C m$^{-2}$ yr$^{-1}$)] | LAI [% (m$^2$ m$^{-2}$)] | NPP [% (g C m$^{-2}$ yr$^{-1}$)] | LAI [% (m$^2$ m$^{-2}$)] |
| Global | 0.20 (0.28) | **−3.05 (−0.06)** | 1.45 (1.96) | **2.54 (0.05)** |
| SWUSA | 12.11 (9.63) | **22.79 (0.21)** | **−16.44 (−13.08)** | −30.08 (−0.28) |
| Amazon | −3.10 (−6.35) | **−6.86 (−0.29)** | 2.58 (5.30) | **2.46 (0.10)** |
| CWAfr | −1.11 (−1.99) | **−13.17 (−0.46)** | 1.26 (2.28) | **4.43 (0.16)** |
| SEAfr | 2.07 (3.16) | 1.54 (0.04) | 1.13 (1.73)) | 3.32 (0.08) |
| India | −3.60 (−3.88) | **−2.28 (−0.03)** | −0.66 (−0.72) | **10.08 (0.15)** |
| SEAsia | −3.66 (−7.85) | **−23.24 (−1.01)** | 4.47 (10.28) | **16.67 (0.72)** |
| NAus | 12.32 (9.66) | 13.81 (0.09) | 40.60 (31.85) | 72.40 (0.49) |

**Table 4.** Absolute and percentage changes in Net Primary Productivity (NPP) and Leaf Area Index (LAI) for El Niño and La Niña scenarios corresponding to Fig. 5. Statistically significant anomalies ($0 < .01$) are in bold.

The model configuration used in this study links atmospheric states with vegetation dynamics. Given that the vegetation is responsive to temperature, solar radiation, and soil moisture, changes in atmospheric states during an ENSO event also modify the state of the vegetation. Fig. 5 shows the spatial distribution of the net primary production (NPP) and leaf area index (LAI) during the base case as well as anomalies in the same parameters during El Niño and La Niña scenarios. Percentage and absolute changes in the NPP and LAI are presented in Table 4. Our the contrasting effects of El Niño and La Niña anomalies on NPP and LAI across different regions. On a global scale during El Niño occurrences, a slight rise (0.20%) in NPP is noted, although this change lacks statistical significance. On a global scale during El Niño occurrences, a slight rise (0.20%) in NPP is

noted, although this change lacks statistical significance.However, LAI experiences a significant decrease (-3.05%), suggesting a reduction in vegetation cover. Regionally, SWUSA exhibits a notable increase in both NPP (12.11%) and LAI (22.79%) during El Niño, while the Amazon region experiences decreases in both NPP (-3.10%) and LAI (-6.86%). In contrast, during La Niña, positive changes are observed, with notable increases in NPP and LAI in NAus.

The biosphere responds on different time scales to atmospheric changes: while the NPP responds nearly instantaneously, the
LAI responds rather slowly (within years to decades), but changes in PFT composition are often very slow as they are the result of decades to hundreds of years of plant dispersal, establishment and mortality (Hickler et al., 2012). Note that in the vegetation model here, dispersal limitations are not included, i.e. if the climate is suitable for a PFT, it can establish immediately. Thus, the model is likely to overestimate range shifts. However, recent observations of increasing drought- and heat-related tree mortality in many areas across the globe(Parmesan et al., 2022) have shown that PFT compositions can also change rapidly if
the dominant trees die. With the intensification of ENSO, we could be moving towards a new quasi-equilibrium in NPP and LAI state where the vegetation state adapt to the new climate. Fig. 6 shows changes in the vegetational fractional coverage of the dominant PFTs over SWUSA, Amazon, SEAsia, and India in the different scenarios of this climate mode. PFT fractional coverage in CEAfr, SEAfr, and NAus exhibited minimal variations and are therefore excluded from Fig. 6.

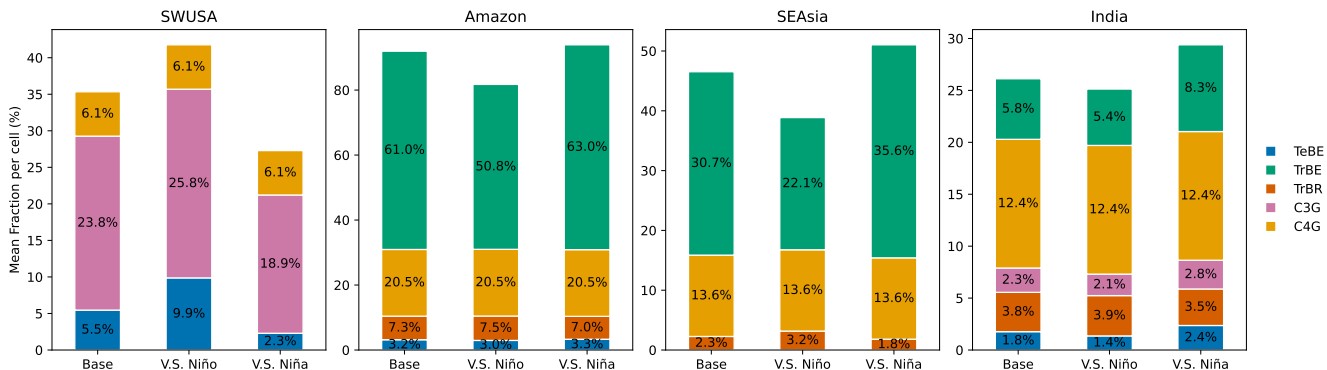

**Figure 6.** Stacked bar plots for the fractional coverage (% of the total area) per PFT during the base case, VS El Niño, and VS La Niña scenarios over the USA, Amazon, Asia, and India. Only PFTs with a fractional coverage higher than 1% are shown. The PFTs shown are: temperate broad-leaved evergreen trees (TeBE), tropical broad-leaved evergreen trees (TrBE), tropical broad-leaved rain-green trees (TrBR), cool C3 grass (C3G), and warm C4 grass (C4G). Based on the average of the last 30 years of 50-year simulations.

The SWUSA is a semi-arid region dominated by grass but also includes temperate broad-leaved evergreen trees (TeBE). In
comparison to the base case, the SWUSA experiences an increase in vegetation during El Niño. TeBE's area coverage expands by 2%, and warm C4 grass (C4G) increases by 4.4%. On the other hand, during La Niña, both TeBE and C4G diminish by 4.9% and 3.4%, respectively. Over the Amazon, there are also notable changes in PFT coverage especially in the El Niño scenario, where tropical broad-leaved trees (TrBE) decrease by 10.2%. On the other hand, TrBE increases by 2% during La Niña. Over SEAsia, during El Niño TrBE decreases by 8.6% during El Niño and increase by 4.9% during La Niña. The area over India
consists of more PFTs and changes compared to the base case mostly occur in the La Niña scenario where the fractional

coverage of TrBE increases by 2.5%. Over CWAfr, SEAfr, NAus, and globally, changes in fractional coverage of different PFTs during the ENSO events were found to be negligible. We observed plant growth expanding into unvegetated territory in scenarios with increased total vegetation. Even in densely forested areas, such as the Amazon region considered in this study, there are patches of land where conditions do not support vegetation. For example, the total vegetated area (including all 12 PFTs) across the Amazon adds up to 92.8% in the base case and 94.8% in the La Niña scenario, meaning that new vegetation was established on 2% of the land area.

### 3.2.3 BVOC emission changes

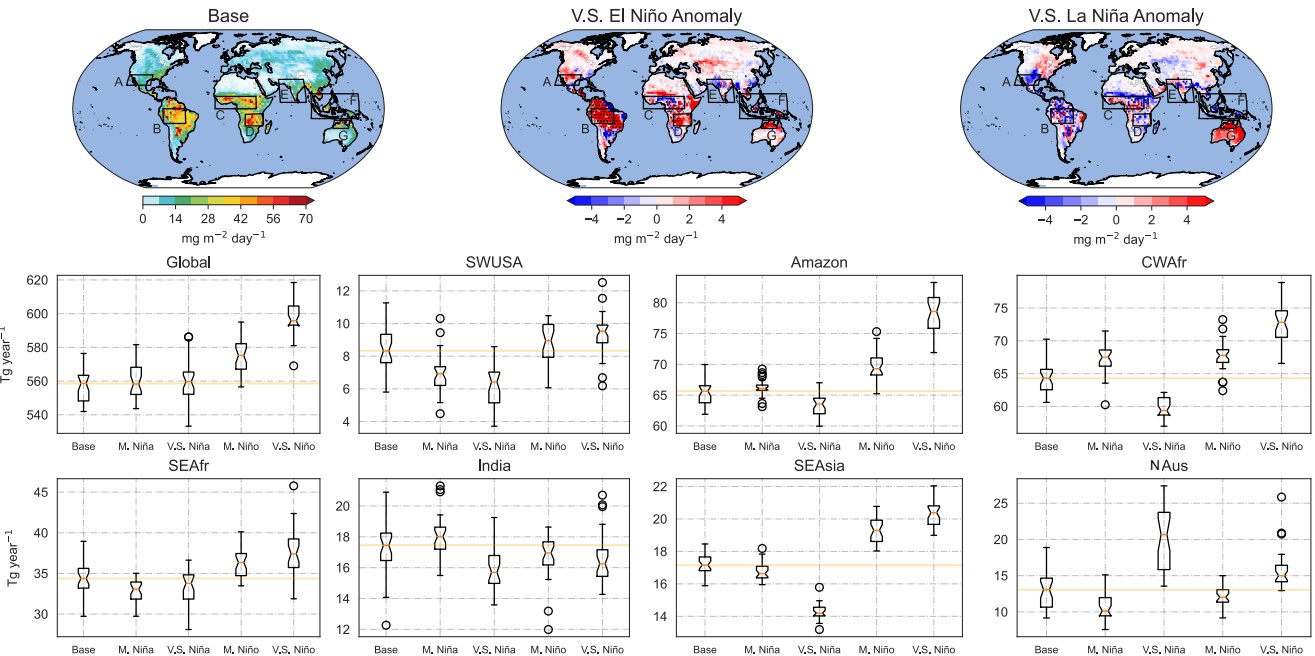

**Figure 7.** Global isoprene fluxes for the base scenarios as well as changes in emissions during Very Strong El Niño and La Niña events. The barplots show emission changes for the different scenarios. The median of the base scenario is indicated in orange.

Fig. 7 shows isoprene fluxes for base conditions as well as emission changes during Very Strong El Niño and La Niña scenarios averaged over 30 years. The box-whisker plots compare emission values for all scenarios over different regions. Global annual isoprene emissions in the base case are 557 Tg. Only El Niño scenarios yield a statistically significant increase in yearly emissions with 573 Tg (+ 2.87%) and 597 Tg (+ 3.95%) for Moderate and Very Strong scenarios, respectively. Over SWUSA, isoprene emissions during the base scenarios are 8.3 Tg yr$^{-1}$ while in the Moderate and Very Strong La Niña emissions are 7 Tg yr$^{-1}$ ($-15.66\%$) and 6.1 Tg yr$^{-1}$ ($-26.51\%$), respectively. Changes during El Niño over SWUSA are statistically insignificant. In the base scenario, emissions over the Amazon are 64.5 Tg yr$^{-1}$, and while anomalies during La Niña are statistically insignificant, isoprene emissions increase to 68.15 Tg yr$^{-1}$ (+ 5.66%) and 77.35 Tg yr$^{-1}$ (+ 19.84%) in Moderate

and Very Strong El Niño scenarios respectively. In CWAfr, anomalies are statistically significant in all scenarios. There is an increase of 2.87% and a decrease of 6.96% in Moderate and Very Strong La Niña scenarios respectively, and an increase of 4.26%, and 11.48% in Moderate and Very Strong El Niño scenarios, respectively.

In SEAfr, statistically significant anomalies only occur in the very strong El Niño scenario with an increase from 34.13 Tg yr$^{-1}$ during the base scenarios to 38.13 Tg yr$^{-1}$ (+11.72%) during the very strong El Niño. Over India, changes in emissions in all scenarios are statistically insignificant. Emissions from SEAsia are 16.56 Tg yr$^{-1}$, and while anomalies for Moderate La Niña are not statistically significant, during Very Strong La Niña emissions drops to 14.99 Tg yr$^{-1}$, and increase to 19.22 Tg yr$^{-1}$ and 19.92 Tg yr$^{-1}$ during Moderate and Very Strong El Niño scenarios, respectively. Lastly, over NAus, base emissions are 12.95 Tg yr$^{-1}$ and increase to 20.14 Tg yr$^{-1}$ during Very Strong La Niña, the only statistically significant anomaly. We again emphasise that these emission changes may be exaggerated as they include long-term changes in the biosphere in this climatic mode. These changes could be seen as an upper range for ENSO-induced emission flux changes in this climatic mode.

**Correlations between fluxes and driving variables**

Fig. 8 displays relationships between the standardized isoprene flux anomalies and surface temperature, radiation, and AI standardized anomalies over land for the different regions during sustained El Niño and La Niña simulations over a 30-year period. Surface temperatures and isoprene fluxes are generally positively correlated during El Niño. The standardized isoprene emissions anomalies correlates well with standardized anomalies for the net solar surface radiation standardized anomaly over the Amazon (0.68), SEAfr (0.47), and SEAsia (0.53) during El Niño, and also during La Niña with $r = 0.81, 0.49$, and 0.68 for Amazon, SEAfr and SEAsia, respectively. This highlights the influence of light availability on photosynthesis and subsequent isoprene production. The isoprene flux anomaly correlates negatively with AI with moderate correlations over the Amazon and SEAfr during El Niño, and over the Amazon, SEAfr, and SEAsia during La Niña, highlighting the link between water availability and isoprene emissions in these regions. Generally, the vegetation activity (both NPP and LAI) positively correlates with isoprene emissions. In both El Niño and La Niña events, the NPP moderately correlates with isoprene fluxes over SEAfr (El Niño $r = 0.51$, La Niña $r = 0.55$), India (El Niño $r = 0.65$, La Niña $r = 0.57$ ), and NAus (El Niño $r = 0.80$, La Niña $r = 0.80$), while moderate positive correlations with the LAI are seen over NWUSA (El Niño $r = 0.58$, La Niña $r = 0.43$), India (El Niño $r = 0.65$, La Niña $r = 0.68$), and NAus (El Niño $r = 0.87$, La Niña $r = 0.86$).

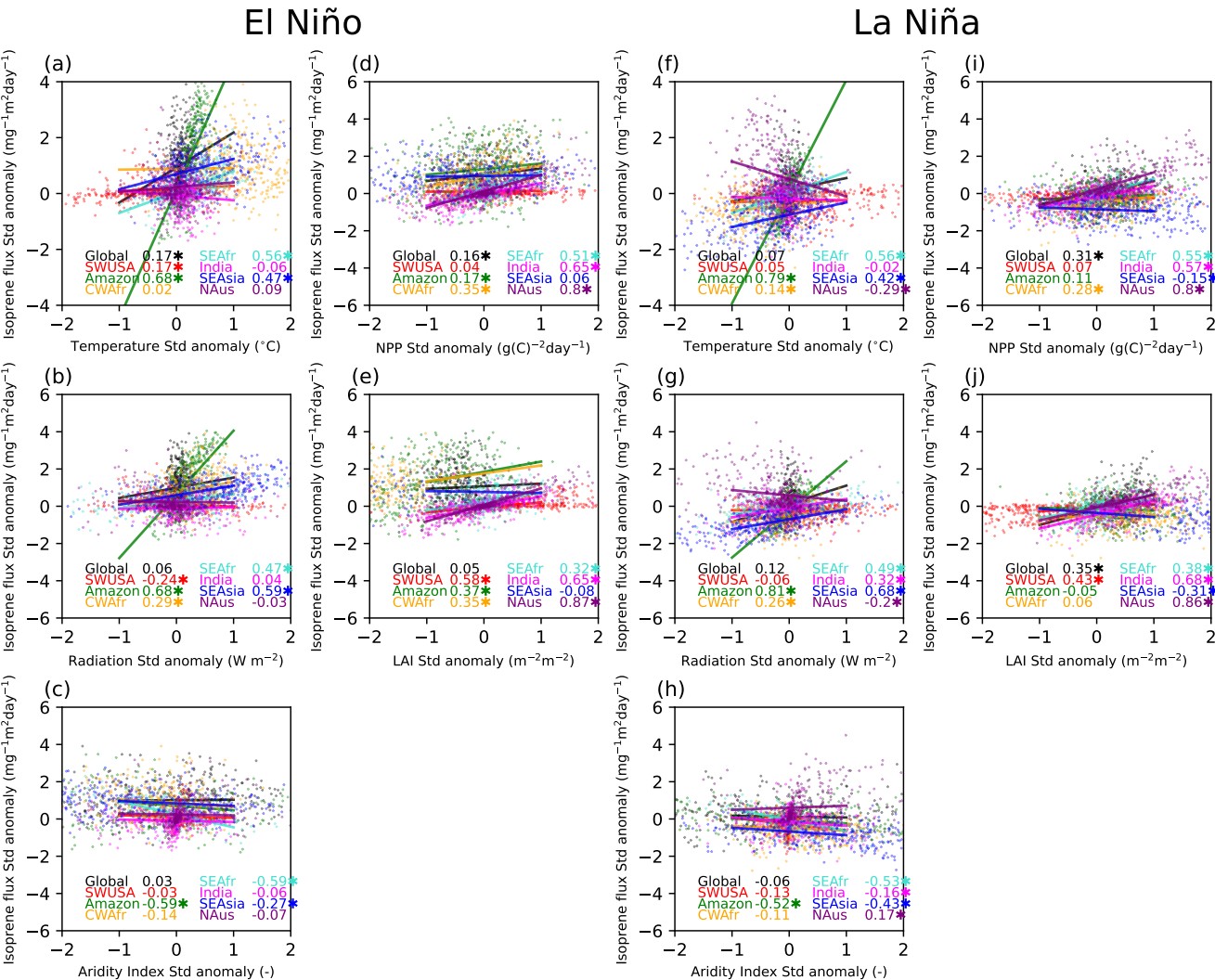

**Figure 8.** Scatter plots with a linear regression fit including the Pearson correlation ($r$) between the standardized isoprene flux anomlaies and standardized temperature, radiation, AI, NPP, and LAI anomalies in different regions for Very Strong El Niño (top panels) and La Niña (bottom panels) events. Correlations with $p < .01$ are marked with a star.

For each grid-cell of the model output, principal component analysis (PCA) was used to determine the variable whose ENSO anomaly correlates most strongly with the BVOC anomaly in order to find the primary cause of the variations in BVOC emissions caused by ENSO. Fig. 9 shows the spatial distribution of the dominant variable that correlates most strongly with changes in isoprene fluxes. BVOC emission changes in the tropics exhibit a predominant influence of temperature, particularly in Africa. However, there are also regions where the flux of BVOCs is primarily driven by surface radiation and anomalies in LAI, most notably in the western part of South America. ENSO-induced BVOC emission anomalies in subtropical regions of

Africa, as well as in the southern United States, Southeast Asia, and Australia seem to be mostly driven by changes in the LAI. In the northern latitudes, particularly across the boreal forest, surface radiation emerges as the primary driver of changes in BVOC emissions.

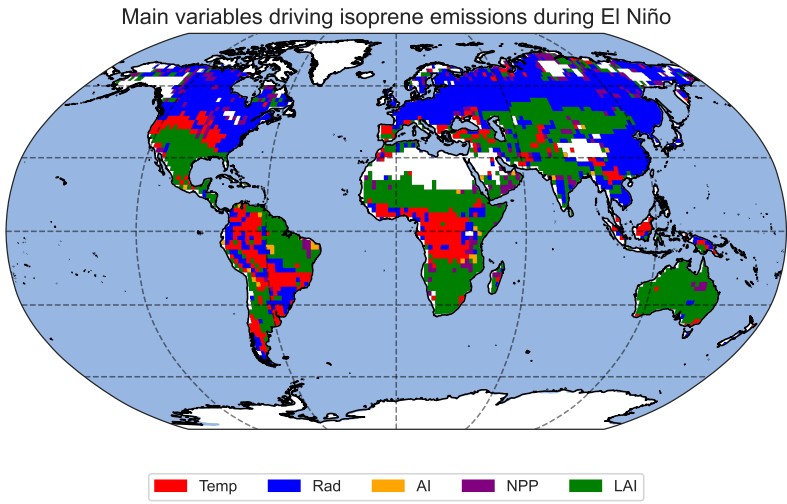

**Figure 9.** Map showing the variables whose anomalies correlates most strongly with the isoprene flux change for every pixel of the model output. The correlations between the isoprene flux and the temperature, radiation, AI, NPP, and LAI anomalies are classified using PCA.

## 4 Discussion

We suggest that the simulations presented in this study are particularly useful as they isolate the response of BVOC emissions resulting from SST/SIC anomalies during ENSO events, and thus avoiding additional forcing from anthropogenic emissions of aerosols and $CO_2$, which are known to influence BVOC fluxes (Heald et al., 2009). However, we acknowledge that these simulations may not fully capture the Earth system's response to ENSO. For instance, our approach simplifies the complexity of the climate system by disregarding the influence of other climate drivers that may interact with ENSO, for example feedbacks via chemistry, SOA, radiation, and aerosol-cloud interactions. The Sustained ENSO simulations also result in inflated BVOC emissions as they also capture long-term changes in the biosphere. However, these simulations are mostly intended for the statistical evaluation of the driving variables response to ENSO and the subsequent changes in BVOC emissions. Furthermore, the simulated vegetation does not include anthropogenic deforestation (and land-use changes), does not include anthropogenic deforestation (and land-use changes), so the simulated vegetation patterns represent potential natural vegetation. EMAC has been shown to have a good representation of the hydrological cycle with a moist bias at the Southern edge of the Himalayas and a dry bias in Amazonia (Xin et al., 2019; Jöckel et al., 2016). This known bias could also lead to amplified water stress affecting BVOC emissions in this region.

While we are confident that the results presented here are relatively robust, we must acknowledge some caveats in our modelling system configuration. Parametriations employed by the model, especially the empirical nature of the BVOC submodels,

might not be able to fully capture all the complexities influencing BVOC emissions from ENSO perturbations. In the study, we also evaluate pure flux emissions from the land biosphere without taking into account changes in atmospheric chemistry and aerosols that may have feedback effects on vegetation and therefore BVOC emissions. For example, recent studies have suggested that SOA from BVOC emission influence surface radiation both directly through scattering, and indirectly via cloud interactions and that higher aerosol scattering is associated with the higher BVOC emissions, while chemical mechanisms also play a role in such feedbacks (Scott et al., 2014; Sporre et al., 2019; Weber et al., 2022).

## 4.1 Isolated ENSO simulations

The BVOC emission anomalies from our isolated simulations agree with previous work linking high BVOC emissions with El Niño years (e.g., 1983, 1987, 1990–1991 and 1994–1995), and lower emissions with La Niña years (e.g 1984–1985, 1988–1989) (Lathiere et al., 2006; Naik et al., 2004). Lathiere et al. (2006) showed that, with $CO_2$ concentrations fixed to 1983, isoprene emissions are higher (1.92%) during El Niño years and lower (−0.63%) during La Niña years compared to the 1983–1995 average. Our simulations, based on 1997/98 (Very Strong El Niño) and 1988/89 (Very Strong La Niña), suggest an increase (2.9%) and a decrease (−0.1%) for El Niño and La Niña, respectively. The disagreement in the magnitudes could arise from differences in the model configurations and the fact that the time frames for "base conditions" differ. Our results suggest that this asymmetry in BVOC emission during El Niño and La Niña arise from regional anomalies within the corresponding ENSO occurrences. However, the observed dissimilarity in asymmetry among various regions implies that this phenomenon is influenced by the distinctive responses of varied vegetation types. While LPJ-GUESS simulates consistent responses in terms of water uptake and water limitation on photosynthesis across all PFTs, the phenological response varies among them. Notably, raingreen trees and grasses, common in savanna ecosystems, tend to "switch off" their leaves at low soil moisture levels to better handle reduced precipitation. This can lead to reduced GPP/NPP, subsequently affecting emissions of BVOCs.

We demonstrate that a single ENSO event has a lasting effect on BVOC emission fluxes, persisting for several years after the event. Even when the ENSO event subsides and normal climatic conditions resume, the biosphere may take time to recover. The prolonged perturbations in BVOC emissions can be attributed to the intricate interplay of various factors, including small changes in microclimate and longer-lasting changes in vegetation. Previous studies have also documented the prolonged impact of ENSO on climatic variables, particularly precipitation, and vegetation dynamics (.eg., Lv et al., 2022; Detsch et al., 2016).

Generally, the correlations during the event years compared to the subsequent two years are stronger for temperature and radiation, but for AI, NPP, and LAI the correlations in the two years following the event are very comparable and in some cases stronger. This indicates that BVOC fluxes respond immediately to anomalies in temperature and radiation but other driving variables such as AI, NPP, and LAI influence the emissions over longer time-scales. It is evident that the correlation patterns vary in different regions during and following El Niño and La Niña events. While temperature and surface radiation seem to play an important role in driving BVOC emissions in most regions, the correlations with the other driving variables show that the biosphere is indeed responding to other signals, emphasising the need for a comprehensive understanding of the underlying mechanisms influencing BVOC emission patterns. This implies also that statistical regression models can hardly capture the processes and dynamics than more complex ecophysiological models like LPJ-GUESS simulate. It is important to note that

this approach has limitations. It simplifies the complexity of the climate system by disregarding the influence of other climate drivers that may interact with ENSO. These drivers can affect the overall climate and may modulate the impact of ENSO on BVOC fluxes in the real world. Therefore, while isolated ENSO simulations provide valuable insights into the ENSO–BVOC relationship, they should be interpreted in the context of the broader climate system and the potential interactions with other drivers.

## 450 4.2 Sustained ENSO simulations

It has been suggested that changes in weather patterns during ENSO events are linked to the rearrangement of the Walker circulation convective centers and teleconnections with midlatitude westerlies (McPhaden et al., 2006; Dai and Wigley, 2000). Our simulations agree with previous studies suggesting that during El Niño the tropics become warmer and drier (Gong and Wang, 1999; Dai and Wigley, 2000), while some areas such as Western North America and East Asia tend to be cooler
and wetter (Ropelewski and Halpert, 1986; Wu et al., 2003). Bastos et al. (2013) investigated the variations in temperature, radiation, and precipitation during El Niño and La Niña events from 2000 to 2011 and revealed significant changes in these variables. Positive temperature anomalies exceeding 1 °C were observed in the Amazon, Central and South Africa, and northern Australia during El Niño, while cooler temperatures were detected in the USA and Europe. In our study, we also observe similar trends, with the strong signal over Australia coming from the significantly cooler temperatures during La Niña. However, we
did not observe such a pronounced influence on European temperatures. Furthermore, our results align with the findings from Bastos et al. (2013) in terms of surface radiation changes. Regarding precipitation, the signal in Bastos et al. (2013) was less distinct, but a decrease in precipitation in the Amazon region during El Niño was suggested. Consistent with these findings, our study revealed higher aridity in the Amazon, supporting the notion of decreased precipitation in this region during El Niño events.

The changes in temperature, surface radiation, and AI during El Niño and La Niña events can have significant impacts on vegetation, as reflected in the changes in NPP and LAI shown in Fig. 5 and Table 4. Our findings are consistent with previous studies linking low global NPP to El Niño years and high global NPP to La Niña years (Zhang et al., 2019; Bastos et al., 2013; Nemani et al., 2003; Behrenfeld et al., 2001). In areas where vegetation growth is at least partly limited by water availability (see Nemani et al. (2003) for global estimates of limiting factors), higher temperatures lead to higher evapotranspiration rates
and increased water stress on vegetation, which may result in reduced NPP and decreased LAI. Cooler temperatures can have varying effects on vegetation, depending on the specific ecological conditions of the region. However, increased surface radiation can enhance photosynthesis and potentially lead to higher NPP. The positive anomalies in NPP observed during La Niña in several regions, such as SWUSA, SEAsia, and NAus, may be attributed to the combined effects of cooler temperatures and increased radiation.

## 475 4.2.1 Global and regional BVOC emission response

BVOC emission changes observed in the sustained ENSO simulations exhibit similar tendencies to those observed in the isolated ENSO simulations. However, the sustained simulations show a stronger signal and larger amplitudes in the anomalies

of BVOC emissions. This enhanced signal can be attributed to the inclusion of long-term changes in the biosphere, as illustrated in Fig. 6. Lathiere et al. (2006) showed stronger correlations between biogenic emissions and the Southern Oscillation Index (defined as the pressure difference between Tahiti and Darwin, which provides a measure of the intensity of El Niño and La Niña episodes) over tropical regions compared to other parts of the world, in particular in the Amazon forest, Southeast Africa, and Southeast Asia. Our findings also show statistically significant changes in isoprene emission fluxes over these regions as shown by the hatched areas in Fig. 7 (and similarly for monoterpenese, see supplementary material).

This simulated positive correlation between BVOC emissions and temperature can be attributed to the enhanced enzymatic activity of plants in higher temperatures (Monson et al., 1992; Sharkey et al., 1996), while the positive correlation with surface radiation could be explained by increased rates of photosynthesis, resulting in enhanced BVOC emissions from vegetation (Sharkey et al., 1996; Harley et al., 1999). It has recently been suggested that limited access to water through soil seems to influence isoprene emissions predominantly through growth stress and, to a lesser extent, closure of stomata, while monoterpene emissions are mostly limited by stomatal closure (Bonn et al., 2019). Similarly, temperature stress also substantially influences BVOC emission fluxes, as can be seen, e.g., by the power law functions of temperature in the description of isoprene emissions (Guenther et al., 2012), e.g., linking a 2 °K temperature anomaly to 23% increased isoprene emissions in China. Areas with higher vegetation productivity tend to exhibit increased isoprene emissions as more carbon resources may be available for isoprene synthesis within actively growing vegetation. Higher LAI values indicate greater foliage density and, consequently, increased potential for isoprene production. The correlations between the driving variables and BVOC anomalies in the regions considered are characterised separately in the following sections:

**South West USA**

In SWUSA, BVOC emissions are higher during El Niño and lower during La Niña. However, the correlations between isoprene emission flux and factors such as temperature, radiation, AI, and NPP are negligible (with a correlation coefficient below 0.29) in both scenarios. Nevertheless, we do observe weak to moderate correlations between isoprene emission flux and LAI in both El Niño and La Niña. Similarly, for monoterpenes, we also find moderate correlations with LAI in both El Niño and La Niña. This suggests that LAI is the primary factor driving BVOC emission fluxes in the Southwestern USA during sustained El Niño and La Niña events.

**Amazon Basin**

In the Amazon region, BVOC emissions are higher during El Niño and lower during La Niña, although the effect during La Niña is relatively weaker compared to El Niño. We have observed that isoprene and monoterpene emission fluxes are moderately to strongly positively correlated with temperature and radiation, and weakly to moderately negatively correlated with AI. However, the correlations with NPP and LAI are weak to negligible. These findings suggest that temperature, radiation, and AI strongly influence isoprene emissions in the Amazon region during both El Niño and, to a lesser extent, La Niña, with temperature and radiation playing particularly significant roles.

**Central West Africa**

In CWAfr, the signal in the BVOC flux anomaly is not very clear. During La Niña, there is an increase in the Moderate scenario and a decrease in the Very Strong scenario. In El Niño, there is a slight decrease in the Moderate scenario and an

increase in the Very Strong scenario. In this regions, correlations with the driving variables was found to be insignificant. Only weak positive correlations between the BVOC emissions and NPP/LAI was found during La Niña. This suggests a potentially
greater influence of vegetation on BVOC emissions in this region.

**South East Africa**

In SEAfr, there are positive BVOC anomalies during El Niño and negative anomalies during La Niña. We have observed weak-to-moderate positive correlations between BVOC emissions and temperature and radiation, as well as weak-to-moderate negative correlations with AI. However, for monoterpenes, the correlation with radiation is negligible. Additionally, we have
found moderate-to-strong positive correlations between isoprene/monoterpene fluxes and NPP. While moderate-to-strong positive correlations exist between monoterpene emissions and LAI, the correlation with isoprene is weak. These correlations suggest that all driving variables are important factors in modulating BVOC emissions in this region.

**India**

The signal in India from our simulations is not very robust. Although there is a slight increase in BVOC emissions during
Moderate La Niña, all other scenarios indicate a decrease in emissions. Regarding the correlations of the BVOC flux anomaly with the driving variables, only NPP and LAI show moderate-to-strong positive correlations. This suggests that BVOC emissions in India during sustained ENSO scenarios are likely to be driven by changes in vegetation.

**South East Asia**

In SEAsia, we have observed a significant increase in isoprene emissions during Very Strong El Niño and a significant
decrease during La Niña. Conversely, monoterpene emissions decrease during El Niño and increase during La Niña, although these changes are not statistically significant. We have found weak-to-moderate correlations between isoprene emissions and temperature (positive), radiation (positive), and AI (negative), while correlations with NPP and LAI are negligible. Monoterpene emissions correlate with temperature and radiation but not with AI, NPP, or LAI. These results indicate that temperature and radiation have a notable impact on BVOC emissions in SEAsia during ENSO.

**North Australia**

In NAus, BVOC emissions experience a slight decrease during Moderate El Niño and La Niña, and an increase during Very Strong El Niño and La Niña, with a stronger signal during La Niña. Both El Niño and La Niña scenarios show weak correlations between BVOC flux and temperature, radiation, and AI. However, strong positive correlations with NPP and LAI are observed during both El Niño and La Niña. These findings suggest that BVOC emissions in NAus are strongly influenced
by NPP and particularly by LAI, regardless of the ENSO phase.

In some regions, e.g., SEAsia (all scenarios except Moderate Niña) and the Amazon (Moderate Niña scenario), see Fig.7 & Fig. S5, we notice asymmetry in the isoprene and monoterpene emissions. The isoprene and monoterpene parameterisations in ONEMIS only differ in the emission factors and the correction factor based on the number of carbon atoms per molecule. This means that the different emission factors can lead to variations in the overall emission rates between the two compounds, even
when other variables are the same. For example, if the emission factor for isoprene is assigned a higher weight compared to monoterpene, the model will amplify the effect of the corresponding variable (e.g., temperature) on isoprene emissions, resulting in a larger increase in isoprene fluxes compared to monoterpene fluxes. Conversely, if the emission factor for monoterpene

is given a higher weight, the model will prioritize the effects of that variable, potentially leading to a larger decrease in monoterpene emissions compared to isoprene emissions.

From our PCA with the driving variables and BVOC fluxes, we conclude that in the Northern latitudes, changes in surface radiation from ENSO correlate best with changes in BVOC emissions, suggesting that in this region BVOC fluxes are mostly influenced by surface radiation anomalies, typically linked with cloud cover and aerosol changes (Scott et al., 2018; Petäjä et al., 2022). Boreal forests, found in higher latitudes, are typically characterised by colder climates with shorter growing seasons. In these regions, the availability of sunlight, represented by surface radiation, plays a crucial role in determining photosynthetic activity and plant growth. Changes in surface radiation, such as alterations in cloud cover or atmospheric conditions, can directly impact the amount of solar energy reaching the vegetation canopy. Increased surface radiation can enhance photosynthesis and, subsequently, BVOC emissions in boreal forests, where plants are sensitive to changes in light availability. However, modelling studies with similar vegetation models suggest that high latitude NPP is primarily limited by cold and short growing seasons, i.e., temperature, not radiation (Lucht et al., 2002; Nemani et al., 2003). This implies that vegetation growth and BVOC controlling factors are not identical. Even though links between ENSO and European precipitation are evident, as pointed out by e.g., Martija-Díez et al. (2023), the effects of the precipitation changes are outweighted by the changes in radiation, resulting in less strong effects compared to the low latitudes. The high dependence of BVOC fluxes on changes in surface radiation over Boreal forests also indicates potential links between emissions of secondary organic aerosol (SOA) precursors, cloud cover, and albedo. This model setup will be used to investigate further the aerosol–cloud–radiation interactions driven by BVOC emissions.

The dependencies highlighted in Fig. 9 are complex but can be associated with the magnitude in the anomalies of the driving variables as well as how different plant species within a specific microclimate respond to such changes. In the tropics, especially in central Africa, changes in isoprene fluxes from ENSO are mostly driven by temperature anomalies. For example, in Central Africa during El Niño, we have strong positive temperature anomalies, but anomalies in surface radiation and aridity are are comparatively smaller (see Fig. 4). This observation potentially elucidates why temperature serves as the primary driver of BVOC anomalies in this particular area. On the other hand, in northeast South America, we observe a substantial impact on temperature, accompanied by significant alterations in surface radiation and AI. These combined effects likely contribute to the robust signal in NPP and, more specifically, to the changes observed in LAI in this region.

Our findings indicate that in the southern USA, northeast South America, South Africa, Central Asia, and Australia, BVOC anomalies are primarily influenced by changes in leaf area index (LAI) resulting from the adaptation of vegetation to new climate states. Although LAI is inherently influenced by atmospheric conditions, the prolonged alterations in LAI - resulting from changes in precipitation patterns, temperature, and radiation regimes associated with sustained ENSO conditions - have a significant impact on BVOC emissions. In the case of monoterpenes (Fig. S7), we see a similar response but we notice that emissions are less driven my radiation anomalies even at higher latitudes. It is also important to note that the changes presented in this study do not account for anthropogenic influences, such as land-use changes, deforestation, and increasing $CO_2$ concentrations which would also influence the response of the biosphere and BVOC emissions.

The results presented here mostly focus on isoprene emission, however, relevant plots for monoterpene emissions can be found in the supplementary material. Monoterpenes' response to the factors driving emissions is similar to that of isoprene, however, the weights of the emission factors are responsible to the different responses. In terms of atmospheric chemistry, oxidation products from monoterpenes are more likely to partition into the particle phase, but are emitted in much less quantities. Nevertheless, some plant types are more likely to emit isoprene whereas others are stronger sources for terpenes.

## 5  Conclusions

This work sheds light on the complex interactions between atmospheric, oceanic, and vegetation processes during ENSO events. Compared to previous work with historical transient simulations (or observations), we used isolated and sustained ENSO scenarios to constrain BVOC emission flux anomalies resulting from ENSO without additional natural forcing from the climate system. The isolated ENSO event simulations suggest a global increase of 2.9% in isoprene emission fluxes over two years with a Very Strong El Niño event. These simulations also show that a single ENSO event perturbs emission fluxes so profoundly that they have not returned to baseline emissions even several years after the event.

We report potential vegetation changes as persistent ENSO conditions shift the vegetation productivity and structure (LAI) into a new quasi-equilibrium state, whereby changes in PFT composition are more transient. Over SE Asia, the fractional coverage of tropical broad-leaved evergreen trees could decreases by 8.6% and increase by 4.9% during persistent El Niño and La Niña scenarios, respectively. Our results show that if ENSO, particularly El Niño, becomes more persistent in the future we should expect an amplification in BVOC flux changes as we enter this new climatic mode with long term influence on the biosphere. In this climate mode, isoprene emissions from the Amazon could increase by 19%. We also show that ENSO-induced variations in BVOC emissions over the tropics are largely related to surface temperature anomalies, whereas surface radiation predominates at higher latitudes. Variations in the LAI were strongly linked to anomalies in BVOC emissions over the subtropics, indicating that ENSO-induced changes in plant phenology are an important driver of BVOC emissions.

In this work we highlight that the impact of ENSO on the Earth system goes beyond changes in temperature and precipitation, but also signficantly impact BVOC emissions, which can have far-reaching implications for the atmosphere and climate. ENSO-induced BVOC emissions changes contribute to the formation of SOA, potentially leading to radiation–climate feedbacks. As BVOC emissions are projected to rise in a warming climate, it becomes imperative to understand and quantify these weather disturbances to accurately predict future BVOC emissions, SOA formation, and their climate feedbacks. Additionally, BVOCs are crucial players in the formation of tropospheric ozone and other harmful air pollutants, posing risks to human health and regional air quality, especially in communities located close to dense forests. Furthermore, BVOCs can act as precursors for greenhouse gases like methane, exacerbating the overall radiative forcing and contributing to climate change. Therefore, it is crucial to understand the intricate interplay between ENSO, BVOC emissions, atmospheric chemistry, and climate for accurately predicting, and mitigating the far-reaching impacts of the ENSO in the climate system.

*Data availability.* The datasets and analysis scripts used in this work are available here: https://doi.org/10.5281/zenodo.8160152

*Author contributions.* RV, AP, and HT conceptualised the study and planned the experiments with significant contributions from JL. RV performed the simulations and data analysis. The results were interpreted by all co-authors, with a special focus on vegetation analysis provided by MF and TH. RV wrote the article with significant inputs from all co-authors.

*Competing interests.* The authors declare that they have no competing interests.

*Acknowledgements.* This research was conducted using the supercomputer Mogon and/or advisory services offered by Johannes Gutenberg University Mainz (https://hpc.uni-mainz.de/, last access: 03 July 2023), which is a member of the AHRP (Alliance for High Performance Computing in Rhineland Palatinate, https: //www.ahrp.info/, last access: 03 July 2023) and the Gauss Alliance e.V. This work was supported by the Max Planck Graduate Center with the Johannes Gutenberg-Universität Mainz (MPGC).

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
