# Peer review of "Changes in BVOC emissions in response to the El Niño-Southern Oscillation"

_EGUsphere, 2023_

## Author Comment (AC1)

Author's response to comments from Anonymous Referee #1:

**"BVOC emission flux response to the El Niño-Southern Oscillation "**

by Ryan Vella et al.

5 We thank editor and referees for taking the time to review our manuscript and for the valuable feedback. Here, the comments from Anonymous Referee #1 (from June 01, 2023) are reproduced in black, while our comments are presented in blue.

**From Anonymous Referee #1's response:**

In this study, the authors analyse potential links between climate variability induced by the El Nino
10 Southern Oscillation and BVOC emission fluxes in a coupled GCM framework with interactive vegetation. They focus on different aspects of ENSO and the associated impacts on the terrestrial biosphere and BVOC emissions. The study is well written and presents interesting results, and I appreciate that the model runs must have been a huge effort to set up. However, I have some concerns that need to be addressed before the manuscript can be accepted for publication and included some general and specific
15 comments below.

Thank you for your positive comments. We acknowledge that the concerns raised here are valid and the recommended amendments greatly improved our manuscript. Please find our detailed responses below.

General comments

20 My major concern is the Results/Discussion section because the discussion is a bit thin. I might have miscounted, but there are only four references to contextualise the results! That's not enough for a discussion and I would like to see a more critical view on the model set up and outcomes in the study. I wonder whether it would help to split the results and discussion into two separate parts of the paper (but this is up to the authors).

25 We agree that the results presented in the submitted manuscript lacked a thorough discussion. The updated version has a separate section for the Discussion. The discussion compares our findings with several studies including the ones cited in the Introduction. More details are provided below, however, we invite the reviewer to refer to the updated manuscript.

Some things that could be discussed are:

30  You used a coupled simulation stressing the importance of land-atmosphere interactions but you don't really dig into describing processes that might influence the BVOC emissions (and how) except for the last sentence in the conclusions.

More details about the model setup and parametrisation of the BVOC emission module used are now provided in Section 2.2. We also mention this in the Discussion section. E.g.:

35  "The positive correlation between BVOC emissions and surface radiation could be explained by increased rates of photosynthesis, resulting in enhanced BVOC emissions from vegetation (Sharkey et al., 1996; Harley et al., 1999). It has recently been suggested that limited soil water access seems to influence isoprene emissions predominantly through growth stress and, to a lesser extent, closure of stomata, while monoterpene emissions are mostly restricted via stomatal closure (Bonn et al., 2019). Similarly,
40  temperature stress also substantially influences BVOC emission fluxes, as can be seen, e.g., by the power law functions of temperature in the description of isoprene emissions (Guenther et al., 2012). Areas with higher vegetation productivity tend to exhibit increased isoprene emissions as more carbon resources may be available for isoprene synthesis within actively growing vegetation. Higher LAI values indicate greater foliage density and, consequently, increased potential for isoprene production."

45  Has your coupled GCM set-up been evaluated and demonstrated to capture BVOC responses sufficiently compared to observations (if observations are available)?

This study builds on the work by Forrest et al. (2020) and Vella et al. (2023); is is now mentioned in the updated manuscript. BVOC flux measurements are generally scarce and it is hard to constrain global budgets from measurements. However, our model setup has been evaluated against several global BVOC
50  emission models where it well-reproduced isoprene and monoterpene emissions. As the current model configuration does not include detailed land-use scenarios, the simulated vegetation corresponds mostly to "natural" vegetation only.

Details on this were presented in Vella et al. (2023), using the submodel ONEMIS (Kerkweg et al., 2006) in our modelling system EMAC (Jöckel et al., 2016) for biogenic emissions:
55  Isoprene
Over the 10-year simulation period considered, the global annual total isoprene fluxes from ONEMIS were found to be 546 Tg yr-1 (standard deviation (SD) = 8 Tg yr-1) with dynamic vegetation and 558 Tg yr-1 (SD = 7 Tg yr-1) with climatological inputs. Jöckel et al. (2016) reported isoprene annual emissions of 488– 624 Tg using ONEMIS, while other studies estimated fluxes of 642 Tg yr-1 (Shim
60  et al., 2005) using 73 prescribed vegetation types, 571 Tg yr-1 (Guenther et al., 2012) using inventories and Olson ecoregion land covers, 467 Tg yr-1 (Arneth et al., 2007) using 10 PFTs from LPJ-GUESS and, more re- cently, 594 Tg yr-1 using 16 PFTs (Sindelarova et al., 2014).

Monoterpenes
Annual totals from ONEMIS were found to be 102 Tg yr-1 (SD = 1 Tg yr-1) with dynamic vegetation inputs and 175 Tg yr-1 (SD = 2 Tg yr-1) with climatological inputs. MEGAN prescribes 54 Tg yr-1 (SD = 0.7 Tg yr-1) and 76 Tg yr-1 (SD = 0.9 Tg yr-1) with dynamic and climatological inputs, respectively. Guenther et al. (2012) gives a global annual monoterpene emission of 157 Tg, while Sindelarova et al. (2014) reported annual total emissions of monoterpenes ranging between 89 and 102 Tg yr-1 over a 30-year simulation period. Arneth et al. (2007) reported 36 Tg yr-1 .

How do other land surface schemes model BVOC emissions and would you expect different results using a different LSM or GCM? Would you expect that your model framework is more suitable to address your research question compared to other coupled models that enable BVOC simulations?

Several Earth System Models (ESMs) and Chemistry-Climate Models (CCMs) employ similar algorithms, such as the Guenther algorithm (Guenther et al., 2012), to simulate Biogenic Volatile Organic Compounds (BVOC) emissions. While some variations exist, these schemes generally yield comparable results, with slight differences in the importance of input parameters, namely radiative fluxes, temperature, soil moisture, and vegetation. Therefore, it is crucial to accurately represent these parameters in the models.

Global scale models generally have a good representation of temperature and radiative fluxes. However, there are larger discrepancies among models when it comes to the hydrological cycle. Nevertheless, the EMAC model has demonstrated a good representation of the hydrological cycle, although it exhibits a moist bias at the southern edge of the Himalayas and a dry bias in Amazonia.

On the other hand, the representation of vegetation parameters (e.g., NPP, Leaf Area Index, leaf density) is less well-described, particularly in chemistry-climate models. Often, these parameters are either imported as climatological or observationally constrained datasets or described using highly simplified parameterizations. To address this weakness, some models have incorporated a dynamic vegetation model within the CCM, allowing for a fully interactive response of vegetation and its influence on BVOC emissions, even under varying climate conditions.

Consequently, our model framework enables the investigation of multiple aspects, extending beyond the scope of the presented study. Various ESMs incorporate atmosphere-land interactions, such as NorESM, EC-Earth, ECHAM, and UKESM. These models utilise the Guenther emission algorithms (e.g., MEGAN or similar) to estimate BVOC emissions. In our recent study (Vella et al., 2023), we demonstrated that our model setup is suitable for investigating land-atmosphere interactions through BVOC emissions. Doubling $CO_2$ sensitivity experiments revealed that BVOC emissions from our model are sensitive to changes in vegetation and temperature. While some models include $CO_2$ inhibition on BVOC emissions (Arneth et al., 2007), the uncertainties surrounding this process remain high, and other studies suggest that $CO_2$ inhibition is not significant compared to the strong dependencies on temperature (Sun et al., 2013). In our setup, $CO_2$ inhibition on BVOC emissions is not included.

There is a wide range of estimates for global isoprene and monoterpene emissions in the literature. A recent study comparing NorESM, EC-Earth, and ECHAM revealed that NorESM has the lowest isoprene emissions at approximately 435 Tg yr$^{-1}$, while EC-Earth and ECHAM exhibit slightly higher emissions at 572 Tg yr$^{-1}$ and 526 Tg yr$^{-1}$, respectively. Regarding monoterpenes, NorESM has the highest global emissions at 118 Tg yr$^{-1}$, followed by EC-Earth at 96 Tg yr$^{-1}$, and then ECHAM at 77 Tg yr$^{-1}$ (Sporre et al., 2020). The models also show significant differences in aerosol radiative effects resulting from BVOC-SOA treatment, primarily due to different parameterizations and SOA treatments employed.

In conclusion, we have confidence in the robustness of our setup to study biosphere-atmosphere interactions arising from ENSO. However, we acknowledge that other model setups may capture these interactions differently. We hope that this study encourages further modeling efforts to evaluate such interactions.

Are there any caveats in the study itself or shortcomings in the model that could have inflated/ underestimated the results?

We highlight the fact that our approach has limitations as it simplifies the complexity of the climate system by disregarding the influence of other climate drivers that may interact with ENSO, for example feedbacks via chemistry, SOA, radiation, and aerosol-cloud interactions. We elaborate more on the fact that the Sustained ENSO simulations result in inflated BVOC emissions as they also capture long-term changes in the biosphere. However, these simulations are mostly intended for the statistical evaluation of the driving variables response to ENSO and the subsequent changes in BVOC emissions. Furthermore, as mentioned above, the simulated vegetation does not include anthropogenic deforestation, such that the simulated vegetation patterns mostly resemble a potentially natural vegetation.

I want to stress that I don't expect detailed answers to all the questions above, they are just some suggestions for potential discussion points.

After reading the discussion and conclusions, it is not clear to me what the implications of the study are. By that I don't mean it the study set-up and results are not sufficiently interesting, but in my view the authors could expand more on the significance of their study in the current climate and future. i.e. if in fact ENSO events do become more sustained and/ or more extreme in a changing climate, you expect increased BVOC emissions. But what does that mean for associated processes in the atmosphere?

Now mentioned in the Conclusion: "As BVOC emissions are projected to rise in a warming climate, it becomes imperative to understand and quantify these disturbances to accurately predict future BVOC emissions, SOA formation, and their climate feedbacks. Additionally, BVOCs are crucial players in the formation of tropospheric ozone and other harmful air pollutants, posing risks to human health and regional air quality, especially in communities located close to dense forests. Furthermore, BVOCs can act as precursors for greenhouse gases like methane, exacerbating the overall radiative forcing and contributing to climate change. Therefore, it is crucial to understand the intricate interplay between

ENSO, BVOC emissions, atmospheric chemistry, and climate for accurately predicting, and mitigating the far-reaching impacts of these ENSO in the climate system."

Your analysis relies on both isoprene and monoterpene emissions but the monoterpene emissions are largely neglected in the manuscript. Do the two types of emissions play different roles in the atmosphere or are they quite similar? You could pick this up in the discussion.

Monoterpenes were mostly not discussed in detail as their response to the factors driving emissions is similar to that of isoprene. In the ONEMIS parameterisations, the only differences between isoprene and monoterpene emission calculations is the emission factors used. We also explain why this could lead to differences, and also asymmetry in the isoprene and monoterpene emissions:

"In some regions, e.g., SEAsia (all scenarios except Moderate Niña) and the Amazon (Moderate Niña scenario), see Fig.7 & Fig. S5, we notice asymmetry in the isoprene and monoterpene emissions. The isoprene and monoterpene parameterisations in ONEMIS only differ in the emission factors and the correction factor based on the number of carbon atoms per molecule. This means that the different emission factors can lead to variations in the overall emission rates between the two compounds, even when other variables are the same. For example, if the emission factor for isoprene is assigned a higher weight compared to monoterpene, the model will amplify the effect of the corresponding variable (e.g., temperature) on isoprene emissions, resulting in a larger increase in isoprene fluxes compared to monoterpene fluxes. Conversely, if the emission factor for monoterpene is given a higher weight, the model will prioritize the effects of that variable, potentially leading to a larger decrease in monoterpene emissions compared to isoprene emissions."

In terms of atmospheric chemistry, oxidation products from monoterpenes are more likely to partition into the particle phase, but are emitted in much smaller quantities. Nevertheless, some plant types are more likely to emit isoprene whereas others are stronger sources of terpenes.

Specific comments

L1: It might be nicer to start with the umbrella term (BVOC emissions which is also in your title), and then divvy it up into isoprene and monoterpene emissions later on? But this is my personal preference and up to the authors.

Text now reads: "Emissions of Biogenic volatile organic compounds (BOVCs) from the terrestrial biosphere play a significant role in major atmospheric processes."

L1: Can you give one example that explains the 'significant role' BVOC emissions play?

Yes, text updated: "BVOCs are highly reactive compounds that influence the atmosphere's oxidation capacity and also serve as precursors for the formation of aerosols that influence global radiation budgets."

L5: ENSO is the most important mode of climate variability and you could state this in the abstract to motivate your study

Now included in the abstract: "It perturbs the natural seasonality of weather systems on both global and regional scales and is considered the most significant driver of climate variability."

175 L35: Typo (?) 'The El Nino-Southern Oscillation (ENSO) is a periodic oscillation'

Typo fixed.

L58: Are there any assumptions that might explain the impact of ENSO on BVOC emissions in higher latitudes?

ENSO does not only affect the low latitudes, but via global teleconnection patterns, also alters the merid-
180 ional temperature gradient and precipitation patterns, even in Europe (Martija-Díez et al., 2023).Conse-
quently, ENSO has to be regarded as a global scale phenomenon, though the impact in the low latitudes is substantially stronger.

With reference to the cited study, Müller et al. (2008) found correlations of ONI with isoprene emissions
185 over higher latitudes (correlation coefficient between ($-0.3$ and $0.3$) especially when a 6-month shift was applied. The study does not go into details on the assumptions that might explain the impact of ENSO on BVOC emissions in higher latitudes, however, BVOC fluxes in higher latitudes are generally low so the impact on anomalies should not be so significant. This is also why in this study we focus on regions close or in the tropics.

190 L83-84: The citations are a bit off - 'aerosol-cloud interactions (e.g. Tost, 2017). In this study, version [...] used in comprehensive model intercomparison studies (e.g. Joeckel et al., 2016)'

The citations used here have been selected to describe the comprehensive modelling system and to show potential of follow-up studies using the interactive BVOC emissions.

L87: Could define LPJ-GUESS in the first line of the section (L86, sorry for being pedantic)

195 Now defined in the first line.

L96: Why did you exclude land-use change? The use of PNV could also be a discussion point

This is a limitation of or current model setup. The new version of LPJ-GUESS will include land use functionality. Now clarified in manuscript.

L98-105: You very superficially describe the different components of the model, fair enough – but given this study is focussed on the BVOC it'd be nice to know if there is one core process or something that describes the BVOC module and how it links land surface and atmosphere in your model set up.

More details about the model coupling and the BVOC emission module were provided in Section 2.2.

"While efforts for a fully coupled configuration are ongoing, in this work, we use the standard EMAC-LPJ-GUESS coupled configuration, where the vegetation in LPJ-GUESS is entirely determined by the EMAC atmospheric state, soil, N deposition, and fluxes (Forrest et al., 2020). After each simulation day EMAC computes the average daily values of 2-meter temperature, net downwards shortwave radiation, and total precipitation and passes these state variables to LPJ-GUESS. Vegetation information (LAI, foliar density, leaf area density distribution, and PFT fractional coverage) from LPJ-GUESS is then fed back to EMAC for the calculation of BVOC emission fluxes using EMAC's BVOC submodules (Vella et al., 2023), namely ONEMIS (Kerkweg et al., 2006) and MEGAN (Guenther et al., 2006). Both ONEMIS and MEGAN are based on the Guenther algorithms (Guenther et al., 1993), where the BVOC emission flux ($F$) is calculated as a function of the foliar density and its vertical distribution ($D$ [kg dry matter m$^{-2}$]), ecosystem-specific emission factors ($\epsilon$), and a non-dimensional activity factor ($\gamma$) that accounts for the photosynthetically active radiation (PAR) and temperature:

$$F = [D]\ [\epsilon]\ [\gamma] \tag{1}$$

In this work, we evaluate fluxes from ONEMIS, which is the standard and most established emission module in EMAC. Emissions are calculated at four distinct canopy layers, which are defined by the leaf area density (LAD) and the leaf area index (LAI). The attenuation of the PAR is determined for each level by considering the direct visible radiation and the zenith angle. Using the proportions of sunlit leaves and the overall biomass, emissions from both sunlit and shaded leaves within the canopy are estimated. Further technical details for canopy processes employed in ONEMIS can be found in Ganzeveld et al. (2002). While validating pure BVOC fluxes from models using observations remains challenging, this setup was evaluated and demonstrated to well-capture global BVOC estimates and responses when compared to other modelling studies (Vella et al., 2023). As described in Eq. 1, BVOC emission calculations in this setup are governed by vegetation states ($D$) from LPJ-GUESS that are largely based on temperature, radiation, and soil moisture. Furthermore, the instantaneous surface radiation and temperature levels ($\gamma$) have a large impact on the emission rates. On the the basis of such model parameterisations, we explore the impact on BVOC emission anomalies by evaluating changes in the surface temperature and radiation, the aridity index (AI), the NPP, and the LAI."

L107: Have you defined AMIPII somewhere?

Now defined.

L111: Are your thresholds defining weak, moderate and strong common practice? I.e. can you support this decision with a citation pointing to other research using the same thresholds?

There is no officially published thresholds, however, these are used by NOAA and also in several publication. Now explained in Section 2.3.

"Even though not officially published, this ONI threshold classification has been used by the National Oceanic and Atmospheric Administration (NOAA) (www.climate.gov/news-features/blogs/enso/united-states-el-niÃśo-impacts-0, last access: 03 July 2023) and also in several research articles (e.g., Jimenez et al., 2021; Abish and Mohanakumar, 2013)."

L125: Not sure I understand the last sentence on the page. Are you saying you chose the seven regions because they are mostly in the tropics which are typically areas with high BVOC emissions (can you include a reference to support this statement)? You could further motivate the choice of regions by mentioning that they conveniently happen to be ENSO hotspots as well (except NE Australia)

This is now mentioned as follows: "The regions considered are hotspots for ENSO (apart from NE Australia) and places with generally high BVOC emissions in the tropics (except from SW USA) (Bastos et al., 2013; Vella et al., 2023; Sindelarova et al., 2014). Additionally, we used the BVOC anomaly distribution maps (Fig. 7) to establish the exact dimensions of the bounding box for regions with relatively consistent BVOC anomalies."

L129: Have you defined the 'base conditions' somewhere?

Now defined in Section 2.3

L132: Does this mean that in the 31st and 32nd year you perturb the atmospheric circulation with your ENSO anomalies?

Yes. Text amended.

L142: Typeo - 'Even though'

Fixed, thank you.

L151: Could you write out Jan and Dec to January and December please:)

Now written in full.

L151: Capital Event?

Fixed.

L167: A bit convoluted.. Maybe something like 'however, following the ENSO perturbation fluxes diverge'

Updated.

L189: The definition of the aridity index belongs in the methods section

Moved to Section Section 2.3.

L190: Throughout your manuscript you're not consistent with italic/ not italic 'base' scenarios/ conditions

Italics were removed completely. Now the scenarios are consistently referred to as "base", "Moderate" and "Very Strong" El Niño/La Niña.

L191: Are r-values the correlation coefficients?

Yes, updated.

Table 2: I like that you give both percentage and absolute changes for temperature in Table 2 to get a sense of magnitude. Can you also include the actual values for change in Radiation and AI in the table? It might make it easier to link the table to Figure 4.

All tables now include absolute an % changes.

Figure 3: I think the figure is very small and it's quite hard to see anything on it. Maybe you could rearrange the panels. There is also a lot of white space at the top that maybe could be trimmed? But maybe I just can't see the datapoints. Could you spell out the abbreviations in the caption too (AI, NPP, LAI)?

The panels were rearranged to make them bigger. Looks better now. Abbreviations in the figure captions are spelled out.

Section 3.2.1. is a description of the results – where is the discussion here? For example, are the anomalies shown in Figure 4 what you expect [. . .]? As I said above, it might be easier to split results and discussion but that is up to you.

This is now discussed in the separate Discussion Section.

"It has been suggested that changes in weather patterns during ENSO events are linked to the rearrangement of the Walker circulation convective centers and teleconnections with midlatitude westerlies (McPhaden et al., 2006; Dai and Wigley, 2000). Our simulations agree with previous studies suggesting that during El Niño the tropics become warmer and drier (Gong and Wang, 1999; Dai and Wigley, 2000), while some areas such as Western North America and East Asia tend to be cooler and wetter (Ropelewski and Halpert, 1986; Wu et al., 2003). Bastos et al. (2013) investigated the variations in temperature, radiation, and precipitation during El Niño and La Niña events from 2000 to 2011 and revealed significant changes in these climatic factors. Positive temperature anomalies exceeding 1 °C were observed in the Amazon, Central and South Africa, and northern Australia during El Niño, while cooler temperatures were detected in the USA and Europe. In our study, we also observed similar trends, with the strong signal over Australia coming from the significantly cooler temperatures during La Niña. However, we did not observe such a pronounced influence on European temperatures. Furthermore, our results align with the findings of Bastos et al. (2013) in terms of surface radiation changes. Regarding precipitation, the signal in Bastos et al. (2013) was less distinct, but a decrease in precipitation in the Amazon region during El Niño was suggested. Consistent with these findings, our study revealed higher aridity in the Amazon, supporting the notion of decreased precipitation in this region during El Niño events."

Figure 4: I appreciate the value of including anomalies over the ocean as the temperature and radiation plots show the typical ENSO anomalies over the ocean quite nicely. However I wonder, given this study is mostly focussed on land processes, whether you would consider to mask the ocean and include a supplementary figure of the SST anomalies to demonstrate that your experiment captures ENSO. Especially for the radiation anomalies, it is quite hard to see what's happening for the majority of the land surface because the colorbar is maxed out to fit to the ocean anomalies. In this figure, I'm surprised that the bottom panels do not show a signal in Australia which pops up as one of the most impacted regions in Figure 5. Does the water limitation signal disappear because you use the aridity index here rather than direct precipitation anomalies? None of the other anomalies seem to able to explain the strong signal.

All figures now updated with an ocean mask to make anomalies on land clearer. The SST plot that depicts ENSO conditions is included in the supplement.

Thanks for noting the abnormal signal over Australia in Fig. 5. LPJ-GUESS tends to assign very tiny, but non-zero, values over regions without vegetation. We usually apply a small threshold to discard such values over desert regions. I checked my code and realised that this threshold was being applied after

the two-tailed Student's t-test. So the "significant correlation" shown was actually coming from these tiny insignificant values. Once the thresholds are applied properly this signal over Australia disappears. See updated Fig. 5.

L219: You use the abbreviation SEASIA here but in Table 2 for example it is SEAsia. I'm not sure whether this happening in other places in the manuscript, but can you make sure you are consistent within the manuscript?

SEASIA is a typo. All consistent now.

Figure 5: I'm a bit surprised about this figure but maybe I'm misreading it. The middle panels contrast vegetation anomalies in an extreme El Nino with that of an extreme La Nina right? The patterns almost look identical, especially for NPP, and I had to zoom in to see that they the top middle and right panel are not identical but show small differences in the hatching. I wonder whether there might have been a mistake in the plot. Typically for an extreme El Nino, you would expect a negative signal at least for parts of Australia due to increased water limitation while for a La Nina it would be positive as you show here. I also would have expected a negative signal in the tropical rainforests in South America and South East Asia. Figure 4 shows a somewhat contrasting signal in the Aridity Index and to some degree in the incoming SW radiation (but it's hard to tell due to the colorbar, see comment above). Can you confirm that Figure 5 indeed shows the 'correct' distribution of anomalies, and if so can you explain the signal given it is quite counter-intuitive?

Fig. 5 shows the spatial distribution of NPP and LAI as well as the El Niño anomaly (El Niño − Base conditions) and La Niña anomaly (La Niña − Base). There was indeed a bug in the script for this plot, where the differences were not calculated properly. The updated plots are coherent with your remarks about El Niño / La Niña effects on vegetation. The colour scheme was also updated to shades of blue and red as this better depicted the anomaly distributions.

L.234-235: Can you rephrase this? Nearly instantaneously and rather quickly sound like quite similar timescales to me. Are you showing the lag in vegetation response somewhere? If so can you point the reader to that information? If this is meant to be a more general discussion point, could you include a reference to support this statement?

Rephrased to "rather slowly". We do not explore vegetation lags in detail here and this a more general comment to emphasise the changes seen in Fig. 6. As the model formulation allows for new establishment of PFTs only at the end of the year (see the LPJ/GUESS description for more details), some changes in vegetation distribution patterns show a longer lag in the corresponding emission driving parameters compared to the direct response of e.g., soil water stress.

L263: I probably just missed it, but where did you mention before that the emission changes may be exaggerated? I think this could be a good discussion point, can you unpack this more?

Emissions from the Sustained simulations may be exaggerated in the sense that they include long term changes in the vegetation. More clearly mentioned now.

L279: What does this mean – high NPP and LAI = high isoprene emissions?

Yes, this means that higher NPP and LAI results in more BVOC emissions.

L271-284: Following this section, you define 'strong correlations' as values greater than 0.4? Often correlation coefficients are split into weak/moderate/strong classes, and values around 0.5 would typically considered moderate. I think you should be more careful with your phrasing here and/or define somewhere where your differentiation is coming from (based on significance?). You do need to be consistent with the actual values of the correlation coefficients though; the ones in written in the text do not always match the ones in the figure (small differences only).

We are now consistent throughout by using the following classification: 0.00-0.29 as negligible, 0.30-0.49 as weak, 0.50-0.69 as moderate, 0.70-0.89 as strong, and $\geq 0.90$ as very strong for positive correlations and similarly for negative correlation between 0 and -1.

L289-295: You found relationships based on a Pearson correlation, but you don't explain why temperature anomalies drive isoprene fluxes in Africa, and LAI in the southern USA, north east South America, South Africa, Central Asia and Australia. Is this a surprising result? Is it what you expected? Do you know why this is emerging from the model?

The dependencies from the correlation analysis are discussed in the Discussion section.

These dependencies are complex but can be associated with the magnitude in the anomalies of the driving variables as well as how different plant species with in a specific microclimate respond to such changes. For example, in Central Africa during El Niño, we have a strong positive temperature anomalies, but anomalies in surface radiation and aridity are not so great (see Fig. 4). This observation potentially elucidates why temperature serves as the primary driver of BVOC anomalies in this particular area. On the other hand, in northeast South America, we observe a substantial impact on temperature, accompanied by significant alterations in surface radiation and AI. These combined effects likely contribute to the robust signal in net primary productivity (NPP) and, more specifically, to the changes observed in leaf area index (LAI) in this region.

Our findings indicate that in the southern USA, northeast South America, South Africa, Central Asia, and Australia, BVOC anomalies are primarily influenced by changes in leaf area index (LAI) resulting from the adaptation of vegetation to new climate states. Although LAI is inherently influenced by atmospheric conditions, the prolonged alterations in LAI resulting from changes in precipitation patterns, temperature, and radiation regimes associated with sustained ENSO conditions have a significant impact

on BVOC emissions. It is important to note that the changes presented in this study do not account for anthropogenic influences, such as land-use changes, deforestation, and increasing $CO_2$ concentrations which would also influence the response of the biosphere and BVOC emissions.

390

Boreal forests, at higher latitudes, are typically characterised by colder climates with shorter growing seasons. In these regions, the availability of sunlight, represented by surface radiation, plays a crucial role in determining photosynthetic activity and plant growth. Changes in surface radiation, such as alterations in cloud cover or atmospheric conditions, can directly impact the amount of solar energy reaching the vegetation canopy. Increased surface radiation can enhance photosynthesis and, subsequently, BVOC emissions in boreal forests, where plants are sensitive to changes in light availability.

L312: Your current data availability statement is not sufficient for Earth System Dynamics. You should at least make your analysis code publicly available. For the time being I'm sure a github link (or similar) will be enough but for publication you will be asked to publish a zenodo link anyway so you might as well get started on that now!

Data and analysis code will be made available on zenodo. Model code could not be made public.

**References**

Abish, B. and Mohanakumar, K.: Absorbing aerosol variability over the Indian subcontinent and its increasing dependence on ENSO, Global and planetary change, 106, 13–19, 2013.

Arneth, A., Miller, P. A., Scholze, M., Hickler, T., Schurgers, G., Smith, B., and Prentice, I. C.: CO2 inhibition of global terrestrial isoprene emissions: Potential implications for atmospheric chemistry, Geophysical Research Letters, 34, 2007.

Bastos, A., Running, S. W., Gouveia, C., and Trigo, R. M.: The global NPP dependence on ENSO: La Niña and the extraordinary year of 2011, Journal of Geophysical Research: Biogeosciences, 118, 1247–1255, 2013.

Bonn, B., Magh, R.-K., Rombach, J., and Kreuzwieser, J.: Biogenic isoprenoid emissions under drought stress: different responses for isoprene and terpenes, Biogeosciences, 16, 4627–4645, 2019.

Dai, A. and Wigley, T.: Global patterns of ENSO-induced precipitation, Geophysical Research Letters, 27, 1283–1286, 2000.

Forrest, M., Tost, H., Lelieveld, J., and Hickler, T.: Including vegetation dynamics in an atmospheric chemistry-enabled general circulation model: linking LPJ-GUESS (v4. 0) with the EMAC modelling system (v2. 53), Geoscientific Model Development, 13, 1285–1309, 2020.

Ganzeveld, L., Lelieveld, J., Dentener, F., Krol, M., Bouwman, A., and Roelofs, G.-J.: Global soil-biogenic NOx emissions and the role of canopy processes, Journal of Geophysical Research: Atmospheres, 107, ACH–9, 2002.

Gong, D. and Wang, S.: Impacts of ENSO on rainfall of global land and China, Chinese Science Bulletin, 44, 852–857, 1999.

Guenther, A., Karl, T., Harley, P., Wiedinmyer, C., Palmer, P. I., and Geron, C.: Estimates of global terrestrial isoprene emissions using MEGAN (Model of Emissions of Gases and Aerosols from Nature), Atmospheric Chemistry and Physics, 6, 3181–3210, 2006.

Guenther, A., Jiang, X., Heald, C. L., Sakulyanontvittaya, T., Duhl, T. a., Emmons, L., and Wang, X.: The Model of Emissions of Gases and Aerosols from Nature version 2.1 (MEGAN2. 1): an extended and updated framework for modeling biogenic emissions, Geoscientific Model Development, 5, 1471–1492, 2012.

Guenther, A. B., Zimmerman, P. R., Harley, P. C., Monson, R. K., and Fall, R.: Isoprene and monoterpene emission rate variability: model evaluations and sensitivity analyses, Journal of Geophysical Research: Atmospheres, 98, 12609–12617, 1993.

Harley, P. C., Monson, R. K., and Lerdau, M. T.: Ecological and evolutionary aspects of isoprene emission from plants, Oecologia, 118, 109–123, 1999.

Jimenez, J. C., Marengo, J. A., Alves, L. M., Sulca, J. C., Takahashi, K., Ferrett, S., and Collins, M.: The role of ENSO flavours and TNA on recent droughts over Amazon forests and the Northeast Brazil region, International Journal of Climatology, 41, 3761–3780, 2021.

Jöckel, P., Tost, H., Pozzer, A., Kunze, M., Kirner, O., Brenninkmeijer, C. A., Brinkop, S., Cai, D. S., Dyroff, C., Eckstein, J., et al.: Earth system chemistry integrated modelling (ESCiMo) with the modular earth submodel system (MESSy) version 2.51, Geoscientific Model Development, 9, 1153–1200, 2016.

Kerkweg, A., Sander, R., Tost, H., and Jöckel, P.: Implementation of prescribed (OFFLEM), calculated (ONLEM), and pseudo-emissions (TNUDGE) of chemical species in the Modular Earth Submodel System (MESSy), Atmospheric Chemistry and Physics, 6, 3603–3609, 2006.

Martija-Díez, M., López-Parages, J., Rodriguez-Fonseca, B., and Losada, T.: The stationary of the ENSO teleconnection in European summer rainfall, Climate Dynamics, 61, 2023.

McPhaden, M. J., Zebiak, S. E., and Glantz, M. H.: ENSO as an integrating concept in earth science, science, 314, 1740–1745, 2006.

Müller, J.-F., Stavrakou, T., Wallens, S., De Smedt, I., Van Roozendael, M., Potosnak, M., Rinne, J., Munger, B., Goldstein, A., and Guenther, A.: Global isoprene emissions estimated using MEGAN, ECMWF analyses and a detailed canopy environment model, Atmospheric Chemistry and Physics, 8, 1329–1341, 2008.

Ropelewski, C. F. and Halpert, M. S.: North American precipitation and temperature patterns associated with the El Niño/Southern Oscillation (ENSO), Monthly Weather Review, 114, 2352–2362, 1986.

Sharkey, T. D., Singsaas, E. L., Vanderveer, P. J., and Geron, C.: Field measurements of isoprene emission from trees in response to temperature and light, Tree physiology, 16, 649–654, 1996.

Shim, C., Wang, Y., Choi, Y., Palmer, P. I., Abbot, D. S., and Chance, K.: Constraining global isoprene emissions with Global Ozone Monitoring Experiment (GOME) formaldehyde column measurements, Journal of Geophysical Research: Atmospheres, 110, 2005.

Sindelarova, K., Granier, C., Bouarar, I., Guenther, A., Tilmes, S., Stavrakou, T., Müller, J.-F., Kuhn, U., Stefani, P., and Knorr, W.: Global data set of biogenic VOC emissions calculated by the MEGAN model over the last 30 years, Atmospheric Chemistry and Physics, 14, 9317–9341, 2014.

Sporre, M. K., Blichner, S. M., Schrödner, R., Karset, I. H., Berntsen, T. K., Van Noije, T., Bergman, T., O'donnell, D., and
455     Makkonen, R.: Large difference in aerosol radiative effects from BVOC-SOA treatment in three Earth system models, Atmospheric
        Chemistry and Physics, 20, 8953–8973, 2020.

Sun, Z., Hüve, K., Vislap, V., and Niinemets, Ü.: Elevated [CO2] magnifies isoprene emissions under heat and improves thermal
        resistance in hybrid aspen, Journal of Experimental Botany, 64, 5509–5523, 2013.

Vella, R., Forrest, M., Lelieveld, J., and Tost, H.: Isoprene and monoterpene simulations using the chemistry-climate model EMAC
460     (v2.55) with interactive vegetation from LPJ-GUESS (v4.0), Geoscientific Model Development, 16, 885–906, 2023.

Wu, R., Hu, Z.-Z., and Kirtman, B. P.: Evolution of ENSO-related rainfall anomalies in East Asia, Journal of Climate, 16, 3742–3758,
        2003.

---

## Author Comment (AC2)

Author's response to comments from Anonymous Referee #2

**"BVOC emission flux response to the El Niño-Southern Oscillation "**

by Ryan Vella et al.

We thank editor and referees for taking the time to review our manuscript and for the valuable feedback. Here, the comments from Anonymous Referee #2 (from June 05, 2023) are reproduced in black, while our comments are presented in blue.

**From Anonymous Referee #2's response:**

In this article the authors explore the impacts of ENSO on modelled emissions of biogenic volatile organic
compounds (BVOCs). The article is well written and this is an interesting study, which is certainly within the scope of the journal, but should only be published after the following comments have been addressed.

My major concern is with regards to Section 3 (Results and Discussion). In the absence of a dedicated "Discussion" section I would have expected to see more analysis and comparison to the wider literature
alongside the presentation of the results. In the Introduction, several studies are cited that have used observations to explore the links between ENSO and the biosphere – how do your model results compare to what they found? This is an interesting study and should be published but without some more context the reader is left to do a lot of work themselves to understand the implications of these results.

Thank you for considering this study for potential publication in BG. We acknowledge the lack of discus-
sion in the submitted version. We now included a dedicated Discussion Section that compares our results with several studies, including those cited in the Introduction. Below is some text from the new Discussion section. We invite the reviewer to check out the updated version of the manuscript for further details.

"The BVOC emission anomalies from our isolated simulations agree with previous work linking high
BVOC emissions with El Niño years (e.g., 1983, 1987, 1990–1991 and 1994–1995), and lower emissions with La Niña years (e.g 1984–1985, 1988–1989) (Lathiere et al., 2006; Naik et al., 2004). With $CO_2$ concentrations fixed to 1983, it was shown that isoprene emissions are higher (1.92%) during El Niño years and lower (−0.63%) during La Niña years compared to the 1983–1995 average (Lathiere et al., 2006). Our simulations, based on 1997/98 (Very Strong El Niño) and 1988/89 (Very Strong La Niña),
suggest an increase (2.9%) and a decrease (−0.1%) for El Niño and La Niña, respectively. The variances could arise from differences in the model configurations and the fact that the time frames for "base conditions" differ."

Minor Comments:

Section 1 (Introduction):

Could you expand slightly on the statement you make about future changes in ENSO: "several studies have suggested the possibility of more persistent ENSO conditions in the future (e.g. Bacer et al., 2016; Cai et al., 2015)" - does this mean more frequent, longer lasting, more extreme etc? You can then come back to this in your later Discussion to help the reader understand the implications of your results.

This statement in the Introduction was updated as follows: "some studies have suggested the possibility
of increased frequency of extreme ENSO events under greenhouse warming (e.g, Cai et al., 2015, 2021)." This means that ENSO evens are expected to occur more often with bigger intensities, but not necessarily longer lasting.

Section 2.2:

Can you justify the use of BVOC fluxes from ONEMIS (rather than MEGAN) if they are the only
emissions used here.

Both ONEMIS and MEGAN could have been used for this study. We decided to use ONEMIS here as this module is the standard and more established emission model in EMAC. ONEMIS has been integrated in EMAC and used for a long time and thus BVOC emsissions from ONEMIS could be compared with previous studies. Additionally, the current MEGAN version in EMAC uses the parameterised canopy
environment emission activity (PCEEA) algorithm (only considering above-canopy photosynthetic photon flux density) rather than the alternative detailed canopy environment model that calculates light and temperature at each canopy depth. On the other hand, in ONEMIS, emissions are calculated within four distinguished layers of the canopy. In Vella et al. (2023), we found some artifacts in the BVOC emissions from MEGAN resulting from the PCEEA. For example, in dense forests, higher LAI (e.g. from
increased temperature) at the top of the canopy could result in more shade in the lower parts of the canopy, resulting in a net decrease in BVOC emssions.

Section 2.3:

The description of the simulation set up for the isolated scenarios is clear in that base conditions are used throughout the 50 years but with an isolated El Nino / La Nina in years 31-32. It would be useful to add
some clarification on what the base conditions are, you mention using the SST/SIC data as forcing data to construct the El Nino / La Nina scenarios but it's not clear what is used for the non El Nino / La Nina years. It is later mentioned in the description of the sustained simulations but needs articulating sooner and in addition to temperature, what time period do the CO2 concentrations represent? Could you also add here clarification of what happens in the year following the isolated El Nino / La Nina.

Thanks for pointing this out. Section 2.3 was updated and now clearly states what we mean by "Base conditions" i.e. SST/SIC average from 1980-2009. Is is also mentioned that we keep $CO_2$ concretions fixed to 348 ppmv, representing the year 2000.

Section 3.1:

It may be beyond the scope of this paper to demonstrate this here but can you be satisfied that your
modelling set up captures the observed relationships between e.g., temperature, radiation and BVOC emission fluxes. You can refer to other studies to support this but at the moment the reader is expected to assume that this is the case.

Our results are in good agreement with most studies out there in the ENSO-induced changes for temperature, radiation, NPP, LAI, and BVOC emissions. This is now discussed in the Discussion Section.

"The changes in temperature, surface radiation, and AI during El Niño and La Niña events can have significant impacts on vegetation, as reflected in the changes in NPP and LAI shown in Fig. 5 and Table 4. Our findings are consistent with previous studies linking low global NPP to El Niño years and high global NPP to La Niña years (Zhang et al., 2019; Bastos et al., 2013; Nemani et al., 2003; Behrenfeld
et al., 2001). Higher temperatures lead to higher evapotranspiration rates and increased water stress on vegetation, which may result in reduced NPP and decreased LAI. Cooler temperatures can have varying effects on vegetation, depending on the specific ecological conditions of the region. However, increased surface radiation can enhance photosynthesis and potentially lead to higher NPP. The positive anomalies in NPP observed during La Niña in several regions, such as SWUSA, SEAsia, and NEAus,
may be attributed to the combined effects of cooler temperatures and increased radiation. "

It would be interesting to understand the difference between the relationships depicted in Figure 3 for the two years during the isolated El Nino / La Nina (green years) and the two years following (yellow years). I.e., which of these variables is driving the change in BVOC emissions once the initial temperature perturbation has gone away, does it change?

We included a new table (Table 2) showing correlations between driving variables and isoprene fluxes: 1) during the event, 2) in the two years following the event, and 3) both time-frames (4 years). This allowed us to discuss in more detail the effect of the driving variables. We found that climatic variables tend to correlate more strongly during the event while the vegetation variables also correlates quite well in the two years following the event, suggesting a longer lasting signal from vegetation.

Section 3.2:

Can you add some clarification to the captions for the Figures and Tables in this section as to the time period that the changes correspond to. From the Methods section I think these must be 30-year means following 20 years of sustained El Nino / La Nina but it would be useful to state that here (especially if my interpretation is not correct!)

Your interpretation is correct. The captions are updated to clarify this point.

In the scenarios that see an increase in total vegetation coverage, do you know which land cover type is being lost? I.e., what is the vegetation expanding into?

After checking the total vegetation coverage in the areas considered, it became clear that there are patches of land missing vegetation (even in areas including dense forest such as the Amazon). As seen in
Fig 6. some PFTs expand while others shrink, however it could also be the case that the total vegetated area increases as it expands into these "empty" patches. A sentence was included to explain this.

Editorial Comments:

Page 2, line 30: correct "oxidant"

Page 3, line 70: correct "us" to "use"

Page 6, line 142: correct "Event"

Page 9, line 192, should "start" be "star"?

Page 12, line 237: should "vegetational" be "vegetation"? (I would change this throughout but could leave for Copernicus Copy Editor's opinion)

Supplement:

Page 1: correct spelling of Table in Fig S1 caption

Manuscript updated accordingly. Thank you.

**References**

Bastos, A., Running, S. W., Gouveia, C., and Trigo, R. M.: The global NPP dependence on ENSO: La Niña and the extraordinary year of 2011, Journal of Geophysical Research: Biogeosciences, 118, 1247–1255, 2013.

Behrenfeld, M. J., Randerson, J. T., McClain, C. R., Feldman, G. C., Los, S. O., Tucker, C. J., Falkowski, P. G., Field, C. B., Frouin, R., Esaias, W. E., et al.: Biospheric primary production during an ENSO transition, Science, 291, 2594–2597, 2001.

Cai, W., Santoso, A., Wang, G., Yeh, S.-W., An, S.-I., Cobb, K. M., Collins, M., Guilyardi, E., Jin, F.-F., Kug, J.-S., et al.: ENSO and greenhouse warming, Nature Climate Change, 5, 849–859, 2015.

Cai, W., Santoso, A., Collins, M., Dewitte, B., Karamperidou, C., Kug, J.-S., Lengaigne, M., McPhaden, M. J., Stuecker, M. F.,
Taschetto, A. S., et al.: Changing El Niño–Southern oscillation in a warming climate, Nature Reviews Earth & Environment, 2, 628–644, 2021.

Lathiere, J., Hauglustaine, D., Friend, A., Noblet-Ducoudré, D., Viovy, N., Folberth, G., et al.: Impact of climate variability and land use changes on global biogenic volatile organic compound emissions, Atmospheric Chemistry and Physics, 6, 2129–2146, 2006.

Naik, V., Delire, C., and Wuebbles, D. J.: Sensitivity of global biogenic isoprenoid emissions to climate variability and atmospheric CO2, Journal of Geophysical Research: Atmospheres, 109, 2004.

Nemani, R. R., Keeling, C. D., Hashimoto, H., Jolly, W. M., Piper, S. C., Tucker, C. J., Myneni, R. B., and Running, S. W.: Climate-driven increases in global terrestrial net primary production from 1982 to 1999, science, 300, 1560–1563, 2003.

Vella, R., Forrest, M., Lelieveld, J., and Tost, H.: Isoprene and monoterpene simulations using the chemistry–climate model EMAC
(v2. 55) with interactive vegetation from LPJ-GUESS (v4. 0), Geoscientific Model Development, 16, 885–906, 2023.

Zhang, Y., Dannenberg, M. P., Hwang, T., and Song, C.: El Niño-Southern Oscillation-induced variability of terrestrial gross primary production during the satellite era, Journal of Geophysical Research: Biogeosciences, 124, 2419–2431, 2019.

---

## Author Comment (AC3)

Author's response to comments from Anonymous Referee #3:

**"BVOC emission flux response to the El Niño-Southern Oscillation "**

by Ryan Vella et al.

5 We thank editor and referees for taking the time to review our manuscript and for the valuable feedback. Here, the comments from Anonymous Referee #3 (from June 05, 2023) are reproduced in black, while our comments are presented in blue.

**From Anonymous Referee #3's response:**

Reviews for "BVOC emission flux response to the El Nino-Southern Oscillation"

10 Isoprene and monoterpene emissions from the terrestrial biosphere play a significant role in major atmospheric processes. Biogenic volatile organic compound (BVOC) emissions are sensitive to climatic influences. This manuscript attempts to understand the relationship between BVOC emission and ENSO events using a global atmospheric chemistry-climate model with enabled interactive vegetation. Overall, the results are reasonable, and I recommend a major revision before acceptance.

15 Many thanks for considering our manuscript for review in BG. Detailed response below.

Major comments:

(1) In Section 2.2 EMAC-LPJ-GUESS configuration, I prefer that you can list some key equations for the parameterizations of BVOC emissions in this study. So we can easily understand why you choose temperature, radiation, AI, NPP, and LAI to investigate their impacts on BVOC emission anomalies.

20 Section 2.1 was extended and now includes more details on the model configuration. Also included the key formula for the BVOC parameterisation in ONEMIS. We explain why we evaluate temperature, radiation, AI, NPP, and LAI to study changes in BVOC emissions.

"While efforts for a fully coupled configuration are ongoing, in this work, we use the standard EMAC-LPJ-GUESS coupled configuration, where the vegetation in LPJ-GUESS is entirely determined by the EMAC
25 atmospheric state, soil, N deposition, and fluxes (Forrest et al., 2020). After each simulation day EMAC computes the average daily values of 2-meter temperature, net downwards shortwave radiation, and total precipitation and passes these state variables to LPJ-GUESS. Vegetation information (LAI, foliar

density, leaf area density distribution, and PFT fractional coverage) from LPJ-GUESS is then fed back to EMAC for the calculation of BVOC emission fluxes using EMAC's BVOC submodules (Vella et al., 2023), namely ONEMIS (Kerkweg et al., 2006) and MEGAN (Guenther et al., 2006). Both ONEMIS and MEGAN are based on the Guenther algorithms (Guenther et al., 1993), where the BVOC emission flux ($F$) is calculated as a function of the foliar density and its vertical distribution ($D$ [kg dry matter m$^{-2}$]), ecosystem-specific emission factors ($\epsilon$), and a non-dimensional activity factor ($\gamma$) that accounts for the photosynthetically active radiation (PAR) and temperature:

$$F = [D]\ [\epsilon]\ [\gamma] \tag{1}$$

In this work, we evaluate fluxes from ONEMIS, which is the standard and most established emission module in EMAC. Emissions are calculated at four distinct canopy layers, which are defined by the leaf area density (LAD) and the leaf area index (LAI). The attenuation of the PAR is determined for each level by considering the direct visible radiation and the zenith angle. Using the proportions of sunlit leaves and the overall biomass, emissions from both sunlit and shaded leaves within the canopy are estimated. Further technical details for canopy processes employed in ONEMIS can be found in Ganzeveld et al. (2002). While validating pure BVOC fluxes from models using observations remains challenging, this setup was evaluated and demonstrated to well-capture global BVOC estimates and responses when compared to other modelling studies (Vella et al., 2023). As described in Eq. 1, BVOC emission calculations in this setup are governed by vegetation states ($D$) from LPJ-GUESS that are largely based on temperature, radiation, and soil moisture. Furthermore, the instantaneous surface radiation and temperature levels ($\gamma$) have a large impact on the emission rates. On the the basis of such model parameterisations, we explore the impact on BVOC emission anomalies by evaluating changes in the surface temperature and radiation, the aridity index (AI), the NPP, and the LAI."

(2) Lots of sentences in the main text should appear in the figure captions. Please revise them through the whole text. For example, Page5 Line117-118, "The base year (ie. The 30-year average SST . . . . Blue (La Nina)." should be placed in the Figure 1 caption. Page 9 Line 191-192 "The r value for each grid is shown and correlations with p < .01 are marked with a start sign" should be placed in the Figure 3 caption.

All figure caption were updated accordingly.

(3) Page 8 Line 176-177 "During El Nino and the subsequent two years, SWUSA experiences a rise of 15.6% and 14.3%, respectively, while a decline of 24.4% is found in SWUSA during the two years following La Nina". The responses seem to be asymmetrical for El Nino and La Nina. So why the response of BVOC to La Nina has the lowest decline in the following two years?

If we look at the driving variables over SWUSA, we see asymmetrical responses during El Niño and La Niña e.g., warmer and cooler surface temperatures, higher NPP/LAI and lower NPP/LAI during El Niño and La Niña, respectively (see Fig. 4 & 5). These changes could explain the response of BVOC emissions

- higher during El Niño because of higher temperatures and elevated NPP/LAI, lower during La Nina because of cooler temperatures and lower NPP/LAI.

65 We added a new table (Table 2) that shows the correlations during and following the event, separately. During La Nina in SWUSA, we see a moderate-to-strong correlations between the isoprene flux and the NPP, LAI, and AI (especially with NPP/LAI in the following 2 years), but not so much with the temperature and srface radiation. Therefore, the response of BVOC to La Niña has the lowest decline in the following two years because of ENSO-induced anomalies in the vegetation states with longer-lasting

70 effects.

Furthermore, even though the emission fluxes depend on several input parameters, their sensitivity across the occurring values is not linear, such that even though La Niña shows cooler temperatures (and therefore not that much heat stress) without significantly reduced NPP and LAI as a direct response,

75 the decline in the emissions is weaker than the increase during El Niño.

(4) English writing need to be improved further.

The manuscript was sent to a professional proofreader. We hope that it reads better now.

Some minors:

(1) Page 1 Line 1: "major atmospheric processes", could you show one or two specific examples.

80 Updated.

(2) Page 2 Line 45-46: "Several studies explored the sensitivity of the terrestrial biosphere to different ENSO phases (e.g. Ahlstrom et al., 2015; Chang et al., 2017; Bastos et al., 2018; Teckentrup et al., 2021)", here is another paper well suitable here. See "Wang, J., Zeng, N., Wang, M., Jiang, F., Chen, J., Friedlingstein, P., Jain, A. K., Jiang, Z., Ju, W., Lienert, S., Nabel, J., Sitch, S., Viovy, N., Wang, H., and

85 Wiltshire, A. J.: Contrasting interannual atmospheric CO2 variabilities and their terrestrial mechanisms for two types of El Ninos, Atmos. Chem. Phys., 18, 10333-10345, 2018."

Citation was added.

(3) Page 6 Line 142: "give realistic insights on changes" => give insights into changes. I think simulated results are not necessarily "realistic".

90 Updated.

(4) Page 10 Line 210: "... anomalies from very strong El Nino and La Nina scenarios", the results in Figure 4 is composite results?

Fig. 4 shows results from the sustained ENSO scenarios. On the left hand-side we show the spatial distribution of the variables in "base conditions" i.e. using climatological SST/SIC (1980-2009), while in the middle and right-hand side we show anomalies by comparing simulations with sustained El Niño / La Niña SST/SIC with the "base" simulation.

The sentence was updated as follows:
"Fig. 4 shows global distributions of surface temperature, net solar radiative flux at the surface, and the AI averaged over 30 years for the base scenario as well as anomalies from Very Strong El Niño (Very Strong El Niño − Base) and La Niña (Very Strong La Niña − Base) scenarios."

(5) Page 13 Line 240-244: Two sentences are duplicate.

Fixed, thank you.

(6) Page 13: "TeBe" => "TeBE"

Updated.

(7) Page 14 Line 262-263: "statistically significant anomalies only occur in the very strong El Niño scenario with and increase from 34.13 Tg yr-1 to 38.13 Tg yr-1 (+11.72%) from base scenarios to very strong El Nino" => statistically significant anomalies only occur in the very strong El Niño scenario with the increase from 34.13 Tg yr-1 during the base scenarios to 38.13 Tg yr-1 (+11.72%) during the very strong El Nino.

Updated.

(8) Figure 8 figure caption: The Person's correlation => The Pearson's correlation

Updated

**References**

115   Forrest, M., Tost, H., Lelieveld, J., and Hickler, T.: Including vegetation dynamics in an atmospheric chemistry-enabled general circulation model: linking LPJ-GUESS (v4. 0) with the EMAC modelling system (v2. 53), Geoscientific Model Development, 13, 1285–1309, 2020.

      Ganzeveld, L., Lelieveld, J., Dentener, F., Krol, M., Bouwman, A., and Roelofs, G.-J.: Global soil-biogenic NOx emissions and the role of canopy processes, Journal of Geophysical Research: Atmospheres, 107, ACH–9, 2002.

120   Guenther, A., Karl, T., Harley, P., Wiedinmyer, C., Palmer, P. I., and Geron, C.: Estimates of global terrestrial isoprene emissions using MEGAN (Model of Emissions of Gases and Aerosols from Nature), Atmospheric Chemistry and Physics, 6, 3181–3210, 2006.

      Guenther, A. B., Zimmerman, P. R., Harley, P. C., Monson, R. K., and Fall, R.: Isoprene and monoterpene emission rate variability: model evaluations and sensitivity analyses, Journal of Geophysical Research: Atmospheres, 98, 12 609–12 617, 1993.

125   Kerkweg, A., Sander, R., Tost, H., and Jöckel, P.: Implementation of prescribed (OFFLEM), calculated (ONLEM), and pseudo-emissions (TNUDGE) of chemical species in the Modular Earth Submodel System (MESSy), Atmospheric Chemistry and Physics, 6, 3603–3609, 2006.

      Vella, R., Forrest, M., Lelieveld, J., and Tost, H.: Isoprene and monoterpene simulations using the chemistry-climate model EMAC (v2.55) with interactive vegetation from LPJ-GUESS (v4.0), Geoscientific Model Development, 16, 885–906, 2023.

---

## Referee Report (RR1)

First I want to apologise for the delay in my responses, I know it's frustrating to wait for feedback. But on to the review: It is quite obvious how much effort went into the revision of the paper which is great and in my opinion really improved the manuscript compared to the version before! However, I still have some (very minor) comments, which I listed below.

L. 1: You have a typo there: BOVCs instead of BVOC

L. 37: At the end of the paragraph I think you could add a sentence stating that BVOC emissions vary between years (with a reference), and this might then link better to the next paragraph where you talk about climate variability associated with ENSO

L. 64: Similar to my suggestion above, make you could highlight the knowledge gap you address in your paper here? I.e. you could talk about uncertainty in ENSO associated BVOC emissions in a future climate.

L. 79-80: Can you rephrase this sentence 'Although global changes in these variables can indicate broad global trends, anomalies associated with the ENSO are often observed at regional scales' . I find the first half of the sentence a bit confusing

L. 83: I know you say this later on in your manuscript, but I think to motivate your introduction you can include here also '[…] North East Australia (NEAus), and these regions are commonly thought to be hotspots for ENSO-associated climate anomalies' or something similar

L. 84 – 94: I think this paragraph would fit better as the second last paragraph? And then after this one you can give an overview about what you do in your study (which you currently do in L. 65 – 84). But of course this is your decision to make, and I'm also happy if you leave it the way it is.

L. 108: Can you define the acronym LPJ-GUESS?

L. 124 – 130: Do I understand correctly that with 'fully-coupled' you mean here that there isn't a feedback from the vegetation to climate variables (temperature, precip, incoming SW radiation?)

L. 144: If soil moisture is such a strong influence on the BVOC emissions, why didn't you use the soil moisture output from LPJ-GUESS but the aridity index instead? To be clear, I'm not asking you to change it, I'm just wondering why.

L. 148 – 155: I think this is great but to me it sounds like it belongs in the discussion. Up to you though!

L. 161: 'greater or equal 0.5° for five' – you're missing a C here

L. 171: Maybe I misunderstood the methods but I thought before you said that the CO2 concentration for 348ppmv is representative of 1983 (here you say 2000)

L. 178: You write here SW USA but in other places it's without a space (i.e. SWUSA)

L. 184: I think you can drop 'with respect to time' here as you state later on you're looking at temporal simulations

L. 183-188: I'm really sorry, I should have thought about this earlier, but I wonder whether you could add a third panel in your methods figure (i.e. Fig. 1) with a schematic that shows the approach you describe in L. 183-188?

L. 196: Instead of 'These simulations' you could also write 'The simulations conducted in this study'

L. 201 – 206: Could also go into the discussion

L. 217 – 220: It is not clear to me what you mean with 'the monthly order of months may be disrupted, meaning that month 1 in the simulation could be March'?

L. 221: Better than what?

Fig. 2 caption: Typo (resposne)

Fig. 3 (and 8) caption: I don't think you're describing everything you show in this figure? You say it's the Pearson correlations but to me it looks like a scatter with a linear regression fitted through the points AND the pearson correlations printed in the figure. Sorry I should have seen this earlier on but only noticed it in this revision!

Table 2: I'm sorry I'm only picking up on this now but two questions: 1. Why did you not include any estimates for significance in this table (which you did for the figures)? 2. Why did you analyse the correlation coefficient between standardised anomalies in Table 2 but in Figure 3 they are not standardised?

L. 259/260: I also wonder about interactions. Again, I'm not asking you to redo this but would you expect different results if you accounted for interactions in your statistical approach (i.e. if you applied partial correlation which allows you to control the effect of other related variables)?

L. 283: Typo: NWAus instead of NEAus

L. 290-292: Do you have any references that support your claim of a potential for either sustained El Nino (or La Nina? It's not clear from your text) in a changing climate?

L. 325: 'indicating a small boost in primary productivity' – here it might be worth noting that this is not statistically significant. In general the NPP changes seem to be not significant in most cases except for the La Nina anomaly for SWUSA? Might be worth mentioning

Fig. 6: Why did you choose these four regions?

L. 383 and 384: Again typo: NWAus instead of NEAus

L. 385: Can you explain your PCA method in the methods section?

L. 404: 'mostly resemble a potentially natural vegetation' – isn't switching LUC off the definition of potential natural vegetation?

L. 405: Do you have a reference for your statement about the dry/ moist biases?

L. 414: I was quite curious about the asymmetry in the response as well when I reread your manuscript. Do I understand it correctly that you suggest that the anomaly magnitude in the climate forcing differs between El Nino and La Nina and that might cause the asymmetric response? Then why is the asymmetry not the same for all regions? Could it be linked to different vegetation types, and/ or does LPJ-GUESS simulate different sensitivities to water stress/ water pulses depending on the underlying vegetation?

L. 470: temperature bias = temperature anomaly?

L. 553 – 560: Great discussion! One thing I wonder: You highlight LAI as a dominant driver for BVOC emissions. Maybe it is different in your model set-up, but if I remember correctly in the 'standard' LPJ-GUESS, LAI does not vary throughout the year for evergreen species (which are dominant in Amazonia and SE Asia) and the carbon allocation happens on an annual timesteps. Do you think this could influence your results?

L. 561: 'can be found in the supplementary'

---

## Author Response (AR2)

Author's response to comments from Anonymous Referee #1 (second round):

**"Changes in BVOC emissions in response to the El Niño-Southern Oscillation"**

by Ryan Vella et al.

5  Once again, we thank editor and referees for taking the time to review our revised manuscript. Here, the comments from Anonymous Referee #1 (from August 07, 2023) are reproduced in black, while our comments are presented in blue.

**From Anonymous Referee #1's response:**

First I want to apologise for the delay in my responses, I know it's frustrating to wait for feedback. But
10  on to the review: It is quite obvious how much effort went into the revision of the paper which is great and in my opinion really improved the manuscript compared to the version before! However, I still have some (very minor) comments, which I listed below.

Thank you for your positive comments. We also understand that many people are on holiday in the summer time and responses could take a bit longer. We truly think that our study benefited a lot from
15  your review! Detailed response below.

L. 1: You have a typo there: BOVCs instead of BVOC

Corrected.

L. 37: At the end of the paragraph I think you could add a sentence stating that BVOC emissions vary between years (with a reference), and this might then link better to the next paragraph where you talk
20  about climate variability associated withuncertainties persist due to the influence of climate change on climate variability and ENSO

The following sentence was added: "BVOC emissions exhibit year-to-year variations attributed to the vegetation's sensitivity to climatic conditions. Our prior investigations have revealed that isoprene fluxes, spanning a decade, display a standard deviation of 8 Tg yr$^{-1}$ (Vella et al., 2023). "

25  L. 64: Similar to my suggestion above, make you could highlight the knowledge gap you address in your paper here? I.e. you could talk about uncertainty in ENSO associated BVOC emissions in a future climate.

This sentence was added a the end of the paragraph: "However, uncertainties persist due to the influence of climate change on ENSO variability."

30  L. 79-80: Can you rephrase this sentence 'Although global changes in these variables can indicate broad global trends, anomalies associated with the ENSO are often observed at regional scales' . I find the first half of the sentence a bit confusing

Sentence modified to: "While changes in these parameters on a global scale can indicate overarching trends, anomalies connected to the ENSO are often identified at a regional level. "

35  L. 83: I know you say this later on in your manuscript, but I think to motivate your introduction you can include here also '[...] North East Australia (NEAus), and these regions are commonly thought to be hotspots for ENSO-associated climate anomalies' or something similar

Text included.

L. 84 – 94: I think this paragraph would fit better as the second last paragraph? And then after this one
40  you can give an overview about what you do in your study (which you currently do in L. 65 – 84). But of course this is your decision to make, and I'm also happy if you leave it the way it is.

Agreed. Updated accordingly

L. 108: Can you define the acronym LPJ-GUESS?

Yes, updated.

45  L. 124 – 130: Do I understand correctly that with 'fully-coupled' you mean here that there isn't a feedback from the vegetation to climate variables (temperature, precip, incoming SW radiation?)

Correct. We use EMAC states to determine the vegetation in LPJ-GUESS but the only feedback going back to EMAC are the interactive BVOC fluxes. Updated as follows: "While efforts towards a fully coupled configuration are ongoing, in this work, we use the standard EMAC-LPJ-GUESS coupled configuration,
50  where the vegetation in LPJ-GUESS is entirely determined by the EMAC atmospheric state, soil, N deposition, and fluxes (Forrest et al., 2020), but there is no feedback from the vegetation to climate variables (e.g., changes in albedo and roughness length)."

L. 144: If soil moisture is such a strong influence on the BVOC emissions, why didn't you use the soil moisture output from LPJ-GUESS but the aridity index instead? To be clear, I'm not asking you to change it, I'm just wondering why.

Initially we used the precipitation, but then we went for the AI as it gives a better representation of the water stresses. Indeed, the soil moisture from LPJ-GUESS could have also been used.

L. 148 – 155: I think this is great but to me it sounds like it belongs in the discussion. Up to you though!

Agreed. Moved to the discussion.

L. 161: 'greater or equal 0.5° for five' – you're missing a C here

Corrected.

L. 171: Maybe I misunderstood the methods but I thought before you said that the CO2 concentration for 348ppmv is representative of 1983 (here you say 2000)

For our study we use 348ppmv representative of 2000. In the introduction I cited a study that have used prescribed $CO_2$ representative of 1983, This study was also mentioned in the discussion and the sentence was reformulated to be clear that this was not our work: "Lathiere et al. (2006) showed that, with $CO_2$ concentrations fixed to 1983, isoprene emissions are higher (1.92%) during El Niño years and lower (−0.63%) during La Niña years compared to the 1983–1995 average".

L. 178: You write here SW USA but in other places it's without a space (i.e. SWUSA)

Corrected.

L. 184: I think you can drop 'with respect to time' here as you state later on you're looking at temporal simulations

'with respect to time' removed.

L. 183-188: I'm really sorry, I should have thought about this earlier, but I wonder whether you could add a third panel in your methods figure (i.e. Fig. 1) with a schematic that shows the approach you describe in L. 183-188?

The lower panel in Fig.1 actually shows the SST used for both isolated and sustained simulations. In the isolated simulations we use SST/SIC shown as "base years", then we use the corresponding SST/SIC from event year 1 2 and switch back to base conditions (exactly as shown in the figure). For the sustained simulations we take 12-month SST/SIC from event year 1 & 2 (corresponding to the particular scenario) and we prescribe these data perpetually in our simulations.

L. 196: Instead of 'These simulations' you could also write 'The simulations conducted in this study'

Corrected.

L. 201 – 206: Could also go into the discussion

Moved to discussion.

L. 217 – 220: It is not clear to me what you mean with 'the monthly order of months may be disrupted, meaning that month 1 in the simulation could be March'?

Text was updated as follows and is hopefully more clear: "The baseline simulation employs SST/SIC spanning from January to December, whereas the ENSO simulations adopt distinct 12-month sequences spanning the event years 1 and 2 (see Fig. 1). In these subsequent simulations, there could be a disruption in the sequential order of months given that the ENSO event occurs over two years and could start in March, April, or May of the first event year. Nevertheless, the annual ENSO cycle remains consistent given that these 12 monthly SST/SIC data are used perpetually. ".

L. 221: Better than what?

Text updated: "Compared to the isolated simulations, the sustained simulations better constrain the correlations between BVOC flux emissions, meteorology, and vegetation changes, and they provide statistical confidence that the characterised perturbations are caused by ENSO rather than other variability attributed to the climate system.".

Fig. 2 caption: Typo (resposne)

Corrected.

Fig. 3 (and 8) caption: I don't think you're describing everything you show in this figure? You say it's the Pearson correlations but to me it looks like a scatter with a linear regression fitted through the points AND the pearson correlations printed in the figure. Sorry I should have seen this earlier on but only noticed it in this revision!

Agreed. Captions for Fig. 3, 8, and for relevant figures in the supplementary material were updated: "Scatter plots with a linear regression fit including the Pearson's correlation ($r$)..."

Table 2: I'm sorry I'm only picking up on this now but two questions: 1. Why did you not include any estimates for significance in this table (which you did for the figures)? 2. Why did you analyse the correlation coefficient between standardised anomalies in Table 2 but in Figure 3 they are not standardised?

1. Correlations with $p < 0.01$ in Table 2 are now in bold.

2 . We used the standard anomalies in both instances as described in the figure captions. We found a typo in the x-axis label for the AI panel. Now it reads "Aridity Index Std anomaly (-)" - also for Fig. 8 and supplementary figures.

L. 259/260: I also wonder about interactions. Again, I'm not asking you to redo this but would you expect different results if you accounted for interactions in your statistical approach (i.e. if you applied partial correlation which allows you to control the effect of other related variables)?

While considering interactions in the analysis could provide valuable insights into complex relationships, we think that our results without interactions remain valid. While interactions could introduce additional nuances, our current findings are robust enough to highlight the relationships between the driving variables and BVOC fluxes.

L. 283: Typo: NWAus instead of NEAus

I found some inconsistencies between NW and NE Australia. I checked a map for Australia and our domain actually falls mostly in the "Northern Territory". This domain was renamed to "North Australia (NAus)" throughout the manuscript.

L. 290-292: Do you have any references that support your claim of a potential for either sustained El Nino (or La Nina? It's not clear from your text) in a changing climate?

We agree the this sentence was not clear and could lead to confusion. The text was updated as follows: "While the current climatic conditions do not align with this scenario, it has been suggested that under the influence of climate change, both El Niño and La Niña could become more intense and prolonged (Cai et al., 2015, 2021). The main objective of the presented simulations is to statistically analyse the response of the driving variables to ENSO, enabling us to examine potential lasting impacts on the biosphere and BVOC emissions. "

135   L. 325: 'indicating a small boost in primary productivity' – here it might be worth noting that this is not statistically significant. In general the NPP changes seem to be not significant in most cases except for the La Nina anomaly for SWUSA? Might be worth mentioning

*Agreed. Text updated: "On a global scale during El Niño occurrences, a slight rise (0.20%) in NPP is noted, although this change lacks statistical significance."*

140   Fig. 6: Why did you choose these four regions?

*This sentence was added: "PFT fractional coverage in CEAfr, SEAfr, and NAus exhibited minimal variations and are therefore excluded from Fig. 6."*

L. 383 and 384: Again typo: NWAus instead of NEAus

*Updated as mentioned above.*

145   L. 385: Can you explain your PCA method in the methods section?

*Yes. We added a new section in the Methods section: "Principal Component Analysis (PCA) was used on climate variables to assess their correlation with isoprene emissions during El Niño events (Section 3.2.3). PCA is a statistical method used to simplify and understand complex datasets by transforming the original variables into a new set of orthogonal (uncorrelated) variables called principal components.*
150   *These components are linear combinations of the original variables that capture the maximum variance in the data. For each pixel of the model output, we perform PCA to extract the first principal component for each driving variable (temperature, radiation, aridity index, NPP, and LAI). The correlation between the first principal component of each driving variable and isoprene/monoterpene emission fluxes is computed for each pixel. These correlation values are then used to rank the variables' importance in driving*
155   *isoprene/monoterpene emissions during El Niño events for each pixel."*

L. 404: 'mostly resemble a potentially natural vegetation' – isn't switching LUC off the definition of potential natural vegetation?

*Correct. Sentence updated to: "...does not include anthropogenic deforestation (and land-use changes), so the simulated vegetation patterns represent potential natural vegetation."*

160   L. 405: Do you have a reference for your statement about the dry/ moist biases?

*Citations added.*

L. 414: I was quite curious about the asymmetry in the response as well when I reread your manuscript. Do I understand it correctly that you suggest that the anomaly magnitude in the climate forcing differs between El Nino and La Nina and that might cause the asymmetric response? Then why is the asymmetry not the same for all regions? Could it be linked to different vegetation types, and/ or does LPJ-GUESS simulate different sensitivities to water stress/ water pulses depending on the underlying vegetation?

Yes we think that this asymmetry also arises from the response of different vegetation types. We added this paragraph in the discussion: "Our results suggest that this asymmetry in BVOC emission during El Niño and La Niña arise from regional anomalies within the corresponding ENSO occurrences. However, the observed dissimilarity in asymmetry among various regions implies that this phenomenon is influenced by the distinctive responses of varied vegetation types. While LPJ-GUESS simulates consistent responses in terms of water uptake and water limitation on photosynthesis across all PFTs, the phenological response varies among them. Notably, raingreen trees and grasses, common in savanna ecosystems, tend to "switch off" their leaves at low soil moisture levels to better handle reduced precipitation. This can lead to reduced GPP/NPP, subsequently affecting emissions of BVOCs."

L. 470: temperature bias = temperature anomaly?

Yes, we now use "anomaly" for consistency.

L. 553 – 560: Great discussion! One thing I wonder: You highlight LAI as a dominant driver for BVOC emissions. Maybe it is different in your model set-up, but if I remember correctly in the 'standard' LPJ-GUESS, LAI does not vary throughout the year for evergreen species (which are dominant in Amazonia and SE Asia) and the carbon allocation happens on an annual timesteps. Do you think this could influence your results?

You are correct in noting that the 'standard' LPJ-GUESS model assumes a fixed LAI throughout the year for evergreen species and employs an annual carbon allocation time step. These assumptions could indeed impact the accuracy of our findings regarding BVOC emissions in regions dominated by evergreen vegetation such as Amazonia and Southeast Asia. The lack of temporal variation in LAI and the use of an annual time step may limit the model's ability to capture the dynamic responses of evergreen species to changing environmental conditions, however, in this study, we focus on the statistical relations using the sustained simulations an therefore the sub-yearly variably is not critical for our analysis.

L. 561: 'can be found in the supplementary'

Corrected.

**References**

Cai, W., Santoso, A., Wang, G., Yeh, S.-W., An, S.-I., Cobb, K. M., Collins, M., Guilyardi, E., Jin, F.-F., Kug, J.-S., et al.: ENSO and greenhouse warming, Nature Climate Change, 5, 849–859, 2015.

Cai, W., Santoso, A., Collins, M., Dewitte, B., Karamperidou, C., Kug, J.-S., Lengaigne, M., McPhaden, M. J., Stuecker, M. F., Taschetto, A. S., et al.: Changing El Niño–Southern oscillation in a warming climate, Nature Reviews Earth & Environment, 2, 628–644, 2021.

Forrest, M., Tost, H., Lelieveld, J., and Hickler, T.: Including vegetation dynamics in an atmospheric chemistry-enabled general circulation model: linking LPJ-GUESS (v4. 0) with the EMAC modelling system (v2. 53), Geoscientific Model Development, 13, 1285–1309, 2020.

Lathiere, J., Hauglustaine, D., Friend, A., Noblet-Ducoudré, D., Viovy, N., Folberth, G., et al.: Impact of climate variability and land use changes on global biogenic volatile organic compound emissions, Atmospheric Chemistry and Physics, 6, 2129–2146, 2006.

Vella, R., Forrest, M., Lelieveld, J., and Tost, H.: Isoprene and monoterpene simulations using the chemistry-climate model EMAC (v2.55) with interactive vegetation from LPJ-GUESS (v4.0), Geoscientific Model Development, 16, 885–906, 2023.